



# HSRL-2 Retrievals of Ocean Surface Wind Speeds

Sanja Dmitrovic[1], Johnathan W. Hair[2], Brian L. Collister[2], Ewan Crosbie[2,3], Marta A. Fenn[2], Richard A. Ferrare[2], David B. Harper[2], Chris A. Hostetler[2], Yongxiang Hu[2], John A. Reagan[4], Claire E. Robinson[2,3,+], Shane T. Seaman[2], Taylor J. Shingler[2], Kenneth L. Thornhill[2,3], Holger Vömel[5], Xubin Zeng[6], Armin Sorooshian[1,6,7]

[1]James C. Wyant College of Optical Sciences, University of Arizona, Tucson, AZ 85721, USA
[2]NASA Langley Research Center, Hampton, VA 23681, USA
[3]Analytical Mechanics Associates, Hampton, VA 23666, USA
[4]Department of Electrical and Computer Engineering, University of Arizona, Tucson, AZ 85721, USA
[5]National Center for Atmospheric Research, Boulder, CO 80307, USA
[6]Department of Hydrology and Atmospheric Sciences, University of Arizona, Tucson, AZ 85721, USA
[7]Department of Chemical and Environmental Engineering, University of Arizona, Tucson, AZ 85721, USA
[+]Deceased

*Correspondence to*: Armin Sorooshian (armin@arizona.edu)

**Abstract.** This study introduces and evaluates ocean surface wind speed retrieval capabilities of the High Spectral Resolution

Lidar – generation 2 (HSRL-2) instrument through comparison with wind speed data collected by National Center for Atmospheric Research (NCAR) Airborne Vertical Atmospheric Profiling System (AVAPS) dropsondes. Wind speed is derived from HSRL-2 measurements of the transmitted laser's specular reflection off the ocean surface. The magnitude of the surface reflectivity is determined by the surface's wave-slope variance, which is driven by surface winds. The assessment relies on the multi-year airborne data set collected as part of NASA's Aerosol Cloud meTeorology Interactions oVer the western ATlantic

Experiment (ACTIVATE) campaign, where HSRL-2 retrievals and AVAPS dropsonde measurements of surface wind speeds were horizontally synchronized owing to their joint deployment on one of two aircraft used during the mission. A total of 577 collocated HSRL-2 - dropsonde surface wind speed data points over the northwest Atlantic Ocean are used for this study. Treating the dropsonde wind speeds as truth, it is found that, through two established wind speed – wave-slope parameterizations, the HSRL-2 wind speed retrievals have small errors (0.15 m s$^{-1}$ ± 1.80 m s$^{-1}$ and 0.62 m s$^{-1}$ ± 1.70 m s$^{-1}$)

and high correlation coefficients (0.89 and 0.88) with dropsonde wind speed measurements. Also, HSRL-2 wind speed error is higher in winter than in summer due at least partly to the higher frequency of low wind speeds and reduced cloud fraction in summer. Two research flights from 28 August 2020 and 1 March 2020 serve as detailed case studies to show the success of the collocation method based on ACTIVATE's spatial-coordination strategy and how HSRL-2 wind speed retrievals can enhance science-oriented studies such as those related to cloud evolution and general air-sea interaction. Another case flight

examined from 11 January 2022 demonstrates the challenge of conducting HSRL-2 wind speed retrievals in high cloud fraction conditions. Overall, this study highlights the airborne HSRL-2's ability to retrieve surface wind speeds with accuracy as well as the potential of using dropsondes to validate aircraft instrument data sets within a field campaign.



## 1 Introduction

The layer between the ocean and free troposphere, known as the marine atmospheric boundary layer (MABL), is the target of
various types of research because of the host of processes that occur, such as the modulation of sensible and latent heat fluxes,
the exchange of gases such as carbon dioxide, cloud evolution, and the transport of aerosol particles (Neukermans et al., 2018).
Improved characterization of MABL dynamics is required to accurately simulate large-scale phenomena such as climate
change and global weather patterns (Paiva et al., 2021). To model these dynamics and associated phenomena, ocean surface
wind speeds are needed. Although not the lone governing factor, surface wind speeds are a critical parameter for modelling
sea salt aerosol emissions from the ocean surface (Reid et al., 2001) and in studies of cloud microphysics (Colón-Robles et al.,
2006) since these winds directly lead to increases in sea salt particle concentrations and activation of these particles that turn
them into cloud condensation nuclei (CCN).

Although lidar systems onboard platforms such as the NASA Cloud-Aerosol Lidar and Infrared Pathfinder Observation
(CALIPSO) satellite have been deployed to measure aerosol and cloud vertical distributions, they also have the capability to
provide horizontally-resolved surface wind speed data (Hu et al., 2008). The underlying principle of the CALIPSO wind speed
retrievals was derived from the theory of Cox and Munk (1954) (referred to as CM54 hereafter), where bidirectional reflectance
measurements of sea-surface glint are used to establish a Gaussian relationship between ocean surface wind speeds and the
distribution of wind-driven wave slopes. CALIPSO emitted laser pulses into the atmosphere and measured the reflectance of
those laser pulses from particles, molecules, and the ocean surface. The magnitude of the measured ocean surface reflectance
was used to estimate the variance of the wave slope distribution (i.e., wave-slope variance). The wind speed ($U$) was then
approximated from the wave-slope variance ($\sigma^2$) through this linear relationship:

$$U = \left(\frac{\langle\sigma^2\rangle - 3.0E-3}{5.12E-3}\right). \tag{1}$$

Although CALIPSO retrievals of surface wind speeds have been used in many studies (e.g., Hu et al., 2008; Josset et al.,
2010a; Josset et al., 2010b; Kiliyanpilakkil and Meskhidze, 2011; Nair and Rajeev, 2014; Murphy and Hu, 2021; Sun et al.,
2023), the main drawback is that one must have an accurate calibration of the measured ocean surface reflectance. This presents
a difficulty for elastic backscatter lidars like CALIPSO, for which the signal is typically calibrated high in the atmosphere
where molecular backscatter dominates and aerosol backscatter is insignificant or can be accurately estimated. The problem
lies in the transfer of this calibration to the ocean surface, which entails accounting for the attenuation of the transmitted and
backscattered light by the intervening atmosphere between the calibration region and the ocean surface. If coincident aerosol
optical depth (AOD) data are available (e.g., from MODIS in the case of CALIPSO) then those AOD data may be used to
estimate the intervening attenuation and transfer the calibration. However, such data are only available during daytime, and
even then are typically not produced in the vicinity of clouds. Estimation of the attenuation from the lidar data alone requires
an assumption of the aerosol extinction-to-backscatter ratio (or "lidar ratio"), so errors in the assumed value can create



significant errors in the estimate of attenuation, especially when AOD is high. Because of this, the wind speed estimates in Hu et al. (2008) were limited to scenes with very low AODs.

This study addresses retrieving wind speed directly from a lidar without other assumptions or external constraints by employing the high-spectral-resolution lidar (HSRL) technique through NASA Langley Research Center's (LaRC's) airborne High Spectral Resolution Lidar – generation 2 (HSRL-2) instrument (Hair et al., 2008; Burton et al, 2018). The HSRL-2 can directly measure vertically resolved aerosol backscatter and extinction profiles without relying on an assumed lidar ratio or other external constraints, enabling accurate estimates of the attenuation of the atmosphere. Therefore, the surface reflectance can be directly determined, providing a measure of the wave-slope variance and thus surface wind speed. This paper focuses on a new ocean surface wind speed product leveraging this instrument's capabilities. This study details the HSRL-2's ocean surface wind speed retrieval methodology and evaluates the aforementioned surface wind speed product through comparison with measurements from National Center for Atmospheric Research (NCAR) Airborne Vertical Atmospheric Profiling System (AVAPS) dropsondes. This work leverages an extensive data set from NASA's Aerosol Cloud meTeorology Interactions oVer the western ATlantic Experiment (ACTIVATE) mission, which had multiple scientific and technological objectives described in detail elsewhere (Sorooshian et al., 2019). The mission featured the joint deployment of the HSRL-2 and dropsonde launcher on one of its two aircraft enabling direct intercomparison between the two instrument data sets. ACTIVATE data across its six deployments between 2020 and 2022 are specifically examined. Section 2 provides a detailed description of the HSRL-2 wind-speed retrieval algorithm, the ACTIVATE mission, the NCAR AVAPS dropsonde system, data collocation procedures, and how statistical analysis on the data is performed. This section also showcases a research flight from 28 August 2020 to demonstrate the success of ACTIVATE's spatially-coordinated flight approach. Section 3 summarizes results of both cumulative wind speed intercomparisons for ACTIVATE's multi-year data set and two additional case study flights highlighting 1) the ability of the HSRL-2 to provide continuous surface wind speed profiles that enable the study of cloud evolution and sea-surface temperature dynamics and 2) the potential drawback of using this wind speed product on days with high cloud fraction conditions. Section 4 presents conclusions discussing the success and limitations of this methodology as well as motivating future studies utilizing aircraft and satellite wind speed data.

## 2 Methods

### 2.1 HSRL-2 Wind Speed Retrieval Method

The NASA LaRC HSRL-2 is a lidar instrument designed for airborne operation with the capability to obtain vertically resolved aerosol properties such as aerosol backscatter and depolarization at three wavelengths (355, 532, 1064 nm) along with aerosol extinction at two wavelengths (355 and 532 nm) (Hair et al., 2008; Burton et al., 2018). In addition to these aerosol products, several new geophysical products have (and continue to be) developed, including an aerosol classification routine (Burton et al. 2012), retrievals of atmospheric mixed layer height (Scarino et al., 2014), ocean subsurface particulate backscatter and



attenuation coefficients (Schulien et al., 2017), cloud optical properties (in development), and the focus of this study, 10-m surface wind speeds. The retrieval method is described in detail below.

As mentioned in Sect. 1, a lidar system emits laser pulses into the atmosphere and the backscattered light from particles
(aerosols) and molecules is collected with a telescope and imaged onto optical detectors where the generated analog electrical signal is digitally sampled as a function of time. Backscatter is also received from the reflection of the laser pulse off the ocean surface and is referred to as the "surface return" signal. To derive surface wind speeds, the surface backscatter ($\beta_{surf}$ , units $sr^{-1}$) is estimated from the surface return signal and related to the wave-slope variance ($\sigma^2$), as detailed in Josset et al. (2010b), through

$$\beta_{surf} = \frac{C_F}{4\pi\sigma^2 cos^5(\theta)} e^{-\frac{tan^2(\theta)}{\sigma^2}}, \tag{2}$$

where $C_F$ is the Fresnel coefficient and $\theta$ is the omnidirectional nadir offset angle. The mean wind speed at 10 m above the sea surface ($U$) is then derived using a piecewise empirical relationship between wind speed and wave-slope variance from Hu et al. (2008) (referred to as Hu08 hereafter), where:

$$U = \left(\frac{\langle\sigma^2\rangle}{0.0146}\right)^2, \langle\sigma^2\rangle < 0.0386, U < 7\ m\ s^{-1}, \tag{3.1}$$

$$U = \left(\frac{\langle\sigma^2\rangle - 3.0E-3}{5.12E-3}\right), 0.0386 \le \langle\sigma^2\rangle < 0.0711,\ 7\ m\ s^{-1} \le U < 13.3\ m\ s^{-1}, \tag{3.2}$$

$$U = 10^{\left(\frac{\langle\sigma^2\rangle + 0.084}{0.138}\right)}, \langle\sigma^2\rangle \ge 0.0711, U \ge 13.3\ m\ s^{-1}. \tag{3.3}$$

The relationships shown in Eqs. 3.1 – 3.3 were established by Hu08 using the comparisons between AMSR-E wind speeds and CALIPSO backscatter reflectance mentioned in Sect. 1. Note that Eq. 3.2 is similar to the linear relationship proposed by
CM54 (Eq. 1) and Eq. 3.3 is similar to the log-linear relationship proposed by Wu (1990). Although there are other relationships that have been provided in the literature such as Venkata and Reagan (2016), the Hu08 relationship is focused on to be consistent with CALIPSO retrievals. Summary statistics are also shown using the CM54 relationship, which is widely applied in current remote sensing retrievals.

With respect to wind speed retrievals, the HSRL-2 instrument offers two major advantages over standard backscatter lidars such as CALIPSO: 1) it can account for atmospheric attenuation between the aircraft and the surface so retrievals can be performed without constraining the retrieval to low optical depth conditions or assuming the lidar ratio, and 2) it has high vertical resolution sampling (1.25 m) that enables accurate correction for ocean subsurface scattering, which makes a small but non-negligible contribution to the measured surface return. The equations for the HSRL-2 532-nm measurement channels
are:

$$P_{mol}(r) = G_{mol}\frac{1}{r^2}F(r)\beta_m^{\parallel}(r)T^2(r), \tag{4.1}$$



$$P_{tot}(r) = G_{mol} G_{i2} \frac{1}{r^2} \left[ \left( \beta_p^{\parallel}(r) + \beta_m^{\parallel}(r) \right) + G_{dep} \left( \beta_p^{\perp}(r) + \beta_m^{\perp}(r) \right) \right] T^2(r) \tag{4.2}$$

where $P_x$ is the total measured signal per sampling interval by the lidar and $r$ denotes the range from the lidar. Here the *mol* subscript denotes the measured signal on the molecular channel, for which all particulate backscatter and the surface return is blocked by an iodine vapor filter. The *tot* subscript denotes the "total" backscatter calculated from the sum of two measurement channels, the co-polarized channel and the cross-polarized channel. These channels are essentially elastic backscatter lidar channels similar to the 532-nm channels on CALIPSO, in that they measure attenuated backscatter from both molecules and particles. The co-polarized channel measures backscatter that is polarized parallel to the linear polarization of the transmitted laser pulses, and the cross-polarized channel measures backscatter with polarization perpendicular to the laser pulses. The volume backscatter coefficient, $\beta$ (units $m^{-1}\ sr^{-1}$), is broken out into components arising from either molecular scattering (*m*) or particulate scattering (*p*) and by polarization parallel ($\parallel$) and perpendicular ($\perp$) to the laser. The combined collection efficiency, optical efficiency, and the overall electronic gain for the signals is denoted by $G_x$. The $T^2$ factor is the two-way transmission of the atmosphere, which accounts for both molecular and particulate scattering and absorption between the lidar and range $r$. A full description of the instrument is described in Hair et al. (2008).

Eqs. 4.1 and 4.2 are generalized such that the backscatter coefficients and transmission factors can be either from the atmosphere or ocean, depending on the altitude (or depth) of the scattering volume. Also, the transmission of the molecular backscatter through the iodine vapor filter, *F*, is based on either the atmosphere (*atm*) or the ocean (*ocn*) scattering regions, as they have different backscatter spectra and thus different iodine filter transmission factors, both of which are determined by laboratory calibrations and the modeled molecular scattering spectra (Hair et al., 2008; Hair et al., 2016). Calibration operations are conducted during each flight to provide the relative gain ratios between the molecular (*mol*) and co-polarized (*par*) channels, $G_{i2}$, and between the co-polarized and cross-polarized (*per*) channels, $G_{dep}$, such that

$$G_{i2} = \frac{G_{par}}{G_{mol}}, \ G_{dep} = \frac{G_{per}}{G_{par}}. \tag{5}$$

After the internal gain ratios (Eq. 5) are applied, the two signals (Eqs. 4.1 and 4.2) have the same relative gain, such that the magnitude of $P_{mol}$ is equal to the magnitude of co-polarized molecular component of $P_{tot}$. As will be shown below, the retrieval implements ratios of these two signals, and therefore neither the absolute gain nor any other absolute calibration factor is required to determine the surface backscatter.

To calculate the surface backscatter, the overall system response must be accounted for. The measured signal ($P$) is the convolution of the normalized system response, ($L$), with the ideal measured signal (i.e., infinite detection bandwidth and delta-function-like laser pulse), this signal being the gain-scaled ($G$), range-scaled ($\frac{1}{r^2}$), attenuated ($T^2$) backscatter coefficient ($\beta$, units $m^{-1}sr^{-1}$), which can be written as





$$P_{ideal}(r) = G \frac{1}{r^2} \beta(r) T^2(r). \tag{6a}$$

$$P(r) = G \int_{-\infty}^{\infty} L(r - \rho) P_{ideal}(\rho) \, d\rho. \tag{6b}$$

The system response includes the impact of the laser's temporal pulse shape, detector response, and analog electronic filter
response.

To account for different scattering media and to better understand how the system response impacts the surface backscatter
calculation, it is helpful to separate the total scattering channel, $P_{tot}(r)$, into three contributions: atmosphere [*atm*], surface
[*surf*], and ocean [*ocn*] as follows:

$$P_{tot}(r) = P_{tot}^{atm}(r) + P_{tot}^{ocn}(r) + P_{tot}^{surf}(r). \tag{7}$$

Using Eq. 6, the last term in Eq. 7, $P_{tot}^{surf}(r)$, can be written as

$$P_{tot}^{surf}(r) = G_{mol}G_{i2} \int_{-\infty}^{\infty} L(r - \rho) \frac{1}{\rho^2} \beta_{surf}\delta(\rho - r_s)T^2(\rho)d\rho \tag{8a}$$

$$P_{tot}^{surf}(r) = G_{mol}G_{i2}L(r - r_s) \frac{1}{r_s^2} \beta_{surf}T^2(r_s) \tag{8b}$$

where the range to the ocean surface is $r_s$ and the volume backscatter coefficient for the ocean surface is represented as
$\beta_{surf}\delta(\rho - r_s)$ (units $m^{-1} \, sr^{-1}$), where $\delta(\rho - r_s)$ is the Dirac delta function centered at $r_s$. The measurement signals as detailed
in Eqs. 6 – 8 are then visualized in Fig. 1.



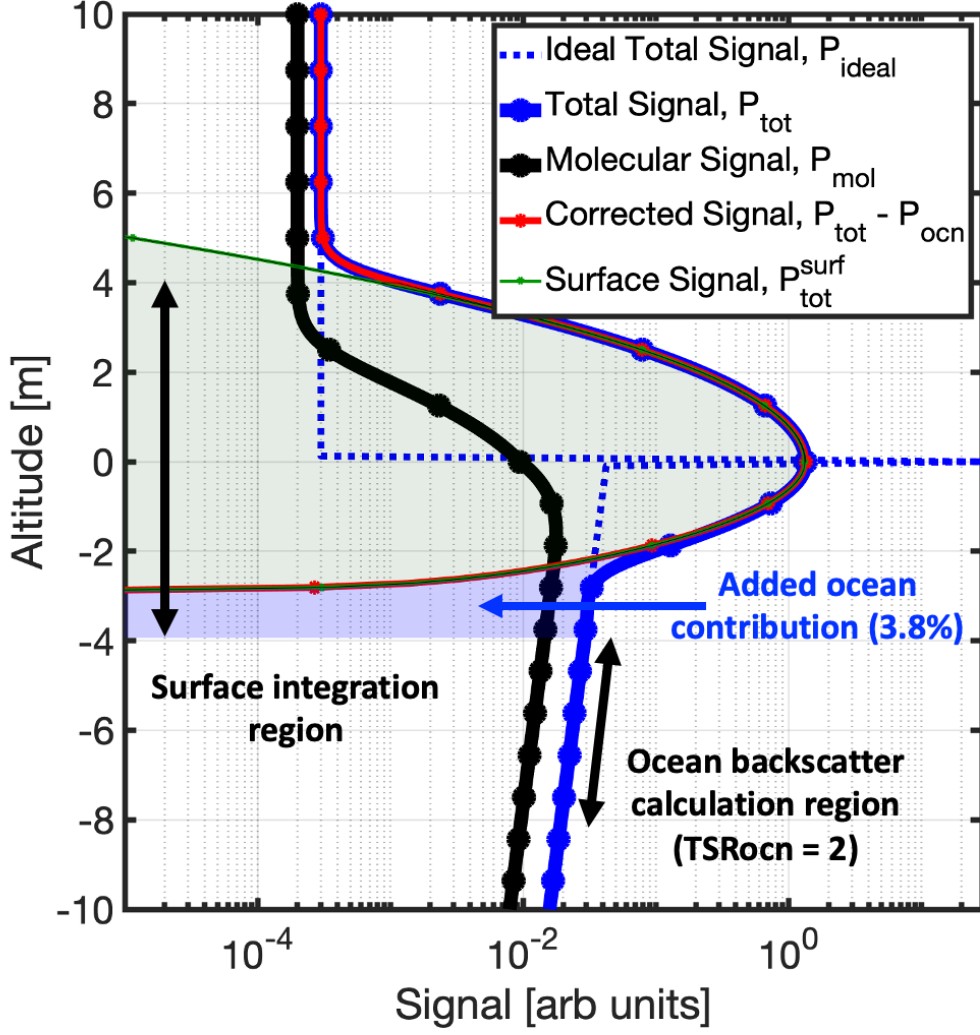

**Figure 1: Visualization of HSRL-2 measurement signals as described in Eqs. 6 – 8. Dashed line denotes ideal total backscatter signal from the atmosphere, surface reflection, and the ocean subsurface. Blue and black lines denote measured signals from total and molecular scattering channels, respectively. Red and green lines show the ocean corrected signal and the ocean surface backscatter, respectively. Dots indicate the altitudes of digitized samples. The sampling rate is 120 MHz, resulting in a vertical spacing of 1.25 m in the atmosphere and 0.94 m in the ocean.**

Figure 1 illustrates the vertical distributions of the measured signals $P_{tot}$ (black) and $P_{mol}$ (blue) along with the $P_{tot}^{surf}$ (green) component of $P_{tot}$. Note that zero altitude is the location of the ocean surface. It is seen from Fig. 1 and Eq. 8b that the surface component $P_{tot}^{surf}$ of the measured signal $P_{tot}$ is not localized to the surface but is instead spread above and below the surface via convolution with the system response function. The atmosphere and ocean components of $P_{tot}$ are also impacted by the convolution as is $P_{mol}$. Rearranging Eq. 8 and integrating the total surface backscatter component over the full vertical extent





of the system response function (i.e., to $\pm\Delta z$), the surface response function can be eliminated in the representation of $\beta_{surf}$ as

shown in Eq. 9.

$$\beta_{surf} = \frac{1}{G_{mol}G_{i2}}\frac{r_s^2}{T^2(r_s)}\int_{r_s-\Delta z}^{r_s+\Delta z}P_{tot}^{surf}(r)\,dr \tag{9}$$

Of course, the measurement that can be accessed is $P_{tot}$, not the surface component $P_{tot}^{surf}$. If $P_{tot}$ were substituted for $P_{tot}^{surf}$

in Eq. 9, $\beta_{surf}$ would be overestimated due to the contribution of ocean subsurface backscatter. The atmospheric contribution

is negligible (i.e., <0.05%) and can be ignored. The magnitude of the contribution of the ocean subsurface scattering depends

on the level of ocean particulate (hydrosol) and as well as molecular seawater backscatter. The magnitude of this scattering

relative to the surface backscatter can impact the retrieved wind speed accuracy. For example, at U = 7 m s$^{-1}$ and assuming

pure seawater (i.e., no hydrosols), the integrated total surface signal would be 5.7% higher than the integrated surface

backscatter if the ocean subsurface scattering is not accounted for in the calculation. This results in a decrease of 0.75 m s$^{-1}$ (-

11% error) in the estimated wind speed. At a 20 m s$^{-1}$ wind speed, the error in the calculated wind speeds results in a decrease

by 2.7 m s$^{-1}$ (-14% error). The correction becomes less as the particulate scattering increases due to increased attenuation of

the ocean subsurface signal, as for the case illustrated in Fig. 1, for which the ocean molecular and particulate scattering are

equal and the integrated subsurface signal is 3.8% higher. The atmospheric signal contribution is much less (~100 times

smaller) than the ocean subsurface signal and therefore its contribution is considered negligible. Fortunately, the HSRL

technique along with the high vertical resolution of the HSRL-2 instrument enables the ocean subsurface contribution to be

estimated. They also enable estimation of the two-way transmittance, $T^2$, and molecular gain factor, $G_{mol}$, in Eq. 9.

For the HSRL-2 instrument, the two-way transmittance is determined directly from the measured molecular channel, $P_{mol}$.

The two-way transmittance to the surface can be estimated as follows,

$$T^2(r_{ns}) = \frac{1}{G_{mol}}\frac{\overline{P_{mol}(r_{ns})r_{ns}^2}}{F(r_{ns})\beta_m^{\|}(r_{ns})}, \tag{10}$$

where $F$ is the iodine vapor filter function (known from lab and in-flight calibration), $\beta_m^{\|}$ is the molecular backscatter coefficient

for the atmosphere (computed from pressure and temperature data from a reanalysis model), and $\overline{P_{mol}(r_{ns})r_{ns}^2}$ is the range-

scaled molecular channel signal near the ocean surface (where $r_{ns}$ is the near-surface range). In practice, this is computed by

averaging data from 60 m to 180 m above the surface. This range is somewhat arbitrary but is chosen as a balance between

ensuring that the signal does not include any of the surface reflectance and low enough to capture most of the attenuation down

to the surface. Substituting Eq. 10 into Eq. 9, one can solve for the surface backscatter,

$$\beta_{surf} = \frac{1}{G_{i2}}\frac{\int_{r_s-\Delta z}^{r_s+\Delta z}r^2 P_{tot}^{surf}(r)dr}{\dfrac{\overline{P_{mol}(r_{ns})r_{ns}^2}}{F(r_{ns})\beta_m^{\|}(r_{ns})}}. \tag{11}$$




To account for the atmosphere and ocean subsurface contributions to the measured signal, Eq. 6 can be rearranged as

$$P_{tot}^{surf}(r) = P_{tot} - P_{tot}^{atm}(r) - P_{tot}^{ocn}(r). \tag{12}$$

Unique to the HSRL-2 instrument, one can use the molecular channel signal to determine the ocean signal near the surface (see Fig. 1). To do so, an estimate of the total ocean scattering ratio (TSR) is employed, which is the ratio of molecular + hydrosol backscatter divided by molecular backscatter. An estimate of the near-surface TSR, $(\overline{TSR_{ocn}})$ is computed using the quotient of the total and molecular channels ($P_{tot}$ / $P_{mol}$) averaged over a small range of depths just below the depth at which the surface signal response goes to zero, as follows:

$$\overline{TSR_{ocn}} \equiv \overline{\left(\frac{\beta_p + \beta_m}{\beta_m}\right)} = \frac{F_{ocn}(r)}{G_{i2}\Delta r} \int_{r_s+\Delta z}^{r_s+2\Delta z} \frac{P_{tot}(r)}{P_{mol}(r)} dr \tag{13}$$

where $F_{ocn}$ accounts for the spectral transmission of the molecular seawater backscatter through the iodine vapor filter and is determined via in-flight and laboratory calibrations. The ocean subsurface component of the total channel backscatter is estimated as follows:

$$P_{tot}^{ocn}(r) = \overline{TSR_{ocn}} G_{i2} \frac{P_{mol}(r)}{F_{ocn}(r)}, below\ the\ surface\ (r > r_s) \tag{14}$$

Here the assumption is that the TSR is constant near the surface. Combining Eqs. 11, 12, and 14 and ignoring the atmospheric contribution $P_{tot}^{atm}$ to the total channel signal, one can compute the absolute surface backscatter using the two measured channels as

$$\beta_{surf} = \frac{\int_{r_s-\Delta z}^{r_s+\Delta z} \left(\frac{P_{tot}(r)}{G_{i2}} - \overline{TSR_{ocn}} * P_{mol}(r)\right) r^2 dr}{\frac{\overline{P_{mol}(r_{ns})r_{ns}^2}}{F_{atm}(r_{ns})\beta_m^{\parallel}(r_{ns})}}. \tag{15}$$

The use of the molecular channel in this way cancels out absolute system gain constants ($G_{mol}$ and $G_{tot}$), provides an estimate of the two-way transmittance of the atmosphere, and enables subtraction of ocean subsurface backscatter. It does not require precise knowledge of the system response function or any other assumptions. With Eq. 15, one can calculate the wave-slope variance through Eq. 2 and then use Eqs. 3.1 – 3.3 to derive surface wind speeds.

## 2.2 ACTIVATE Mission Description

An ideal data set to assess the HSRL-2 ocean surface wind speed product comes from ACTIVATE, which is a NASA Earth Venture Suborbital-3 (EVS-3) mission. The primary aim of ACTIVATE is to improve knowledge of aerosol-cloud-meteorology interactions, which are linked to the highest uncertainty among components contributing to total anthropogenic radiative forcing (Bellouin et al., 2020). There are three major scientific objectives: (i) characterize interrelationships between aerosol particle number concentration ($N_a$), CCN concentration, and cloud drop number concentration ($N_d$) with the goal of decreasing uncertainty in model parameterizations of droplet activation; (ii) advance process-level knowledge and simulation of cloud microphysical and macrophysical properties, including the coupling of aerosol effects on clouds and cloud effects on





aerosol particles; and (iii) assess remote sensing capabilities to retrieve geophysical variables related to aerosol-cloud interactions. This study focuses on the third objective, which has already received attention with ACTIVATE data for retrievals other than ocean surface wind speeds (Chemyakin et al., 2023; Schlosser et al., 2022; Ferrare et al., 2023; Van Diedenhoven et al., 2022). ACTIVATE built a high volume of flight data statistics over the Western North Atlantic Ocean (WNAO) by flying six deployments across three years (2020 – 2022), with a winter and summer deployment each year (Sorooshian et al.,

2023). Winter deployments included the following date ranges: 14 February – 12 March (2020), 27 January – 2 April (2021), 30 November 2021 – 29 March (2022). Summer deployments were as follows: 13 August – 30 September (2020), 13 May – 30 June (2021), 3 May – 18 June (2022). Across all three years, 90 King Air flights during the winter deployment were performed with 373 dropsondes launched while 78 flights during the summer deployment took place with 412 dropsondes launched.


Two NASA Langley aircraft flew in spatial and temporal coordination for the majority of the total flights (162 of 179). A "stacked" flight strategy was developed where a low-flying aircraft collected in situ data in and just above the MABL while a high-flying aircraft simultaneously provided remote sensing retrievals well above the MABL. In doing so, the stacked aircraft would simultaneously obtain data relevant to aerosol-cloud-meteorology interactions in the same column of the atmosphere

and provide a complete picture of the lower troposphere (Sorooshian et al., 2019). In situ measurements of gases, particles, meteorological variables, and cloud properties were conducted by a HU-25 Falcon flying in and above the MABL (< 5 km). The high-flying aircraft (~9 km) was a King Air whose payload included the NASA Goddard Institute for Space Studies (GISS) Research Scanning Polarimeter (RSP) and the two instruments relevant to this work: the NASA LaRC HSRL-2 and the NCAR AVAPS dropsondes (NCAR, 1993). An advantage of the joint deployment of HSRL-2 and AVAPS dropsondes on the King

Air is that the data are spatially synchronized at launch, with wind drift of the dropsondes during descent accounted for with procedures summarized in Sect. 2.3.

The rationale to fly over the WNAO in different seasons was to collect data across a wide range of aerosol and meteorological regimes, with the latter promoting a broad range of cloud conditions (Painemal et al., 2021). A significant meteorological

feature is the North Atlantic Oscillation, which is the oscillation between the Bermuda-Azores High (high pressure system) and the Icelandic Low (low pressure system) (Lamb and Peppler, 1987). In the summer, the Bermuda-Azores High is at its peak and introduces easterly and southwesterly trade winds (Sorooshian et al., 2020). Starting in the fall, the Icelandic Low becomes prominent and introduces westerly winds in the boundary layer. The balancing act between these pressure systems dictates the climate of the North Atlantic and the prevailing transport processes (Li et al., 2002; Christoudias et al., 2012;

Creilson et al., 2003). These transport processes that vary seasonally explain why winter flights coincided with more offshore (westerly) flow containing aerosol types impacted by anthropogenic influence (e.g., Corral et al., 2022), whereas summer flights included more influence from wildfire emissions and African dust among other sources both natural and anthropogenic in nature (Mardi et al., 2021; Aldhaif et al., 2020). Winds and turbulence tend to be stronger in the winter due to higher



temperature gradients between the air and the ocean (Brunke et al., 2022), resulting in a higher fraction of available aerosol
particles in the MABL that activate into cloud droplets in winter coinciding with cold air outbreaks as compared to summer
(Dadashazar et al., 2021; Kirschler et al., 2022; Kirschler et al., 2023; Painemal et al., 2023).

## 2.3 Dropsondes

The AVAPS system utilized the newer, more reliable NRD41 mini sondes for the ACTIVATE mission. In addition to being
smaller, their launching hardware was updated to increase reliability for launches. A variable number of dropsondes were
launched per flight, usually 3 to 4 for routine flights, with more being launched for specific targeted flight opportunities. With
response times much less than 1 second, AVAPS measures position, wind speed (with 0.5 m s$^{-1}$ uncertainty) (Vömel and
Dunion, 2023), and state variables such as pressure, temperature, and humidity, with these data being sampled all the way to
~6 m above the ocean surface. The data are then post processed via NCAR's Atmospheric Sounding Processing Environment
(ASPEN) software where any spurious data are removed including any data returned from the ocean surface itself (Martin and
Suhr, 2021). More details on the AVAPS system and its usage on other aircraft and missions can be found in Vömel et al.
(2021) and Vömel (in review). Not many studies exist on wind speed validation of aircraft instruments with dropsondes (Bedka
et al., 2021), so this study also highlights the potential of using dropsondes to validate aircraft wind speed data.

Because this study focuses on HSRL-2 retrievals of ocean surface wind speed data, dropsonde wind speed measurements
closest to the 10 m neutral stability height are the primary focus from AVAPS. A map of all 577 dropsondes used in this study
from 2020 to 2022 is shown in Fig. 2.



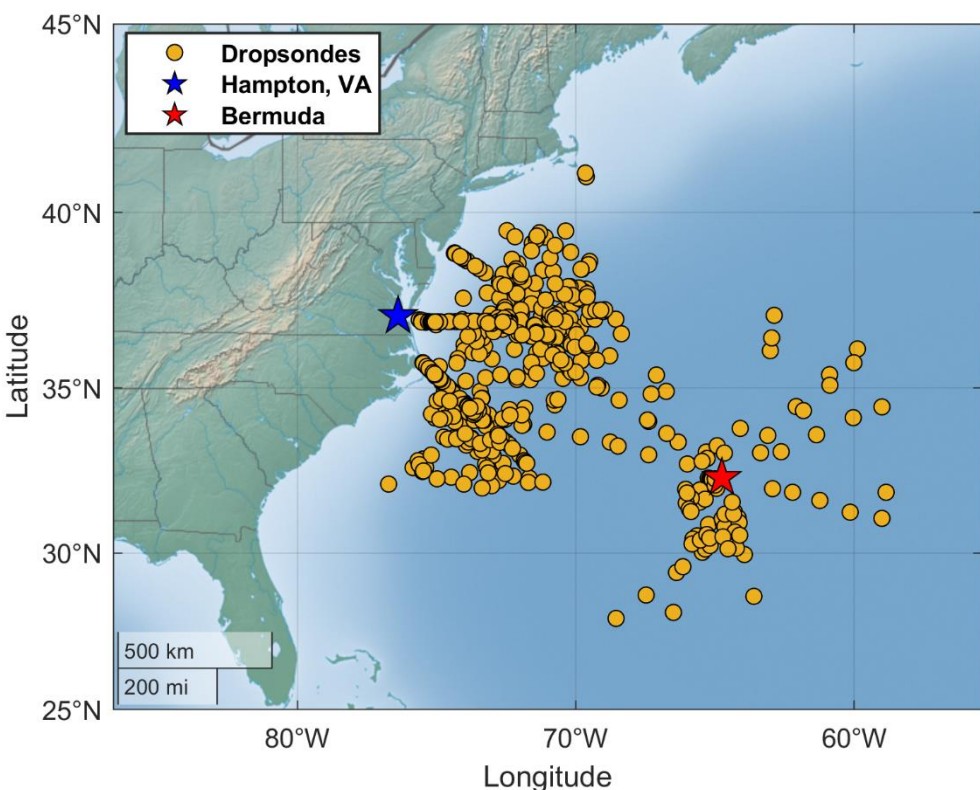

**Figure 2: Map of 577 ACTIVATE dropsondes launched from the King Air between 2020 and 2022 that are used in this wind speed intercomparison study.**

**2.4 Data Collocation Procedure**

A description of the collocation procedure is provided to describe how the wind speed data intercomparisons between the dropsondes and the HSRL-2 are performed. Since wind speeds at the surface are the focus of this study, the dropsonde wind speed data points closest to 10 m (altitude of 11.56 m ± 3.19 m for the 577 points) above sea level are recorded for each launch (multiple launches per flight). Since one data point was taken per dropsonde for each flight, there are 160 recorded dropsonde

measurements for 2020, 245 measurements for 2021, and 335 measurements for 2022. Then, the HSRL-2 wind speed retrieval closest in space and time to the dropsonde wind speed measurement is recorded. Note that there are two HSRL-2 data sets collocated with the dropsonde measurements, one calculated from the Hu08 model and the other from the CM54 model, to allow for comparison of these parameterizations in both the case studies and cumulative results detailed in Sects. 2 and 3. Collocation between the HSRL-2 and the dropsondes is constrained to below 30 km horizontally and below 15 minutes

temporally to remove outliers while preserving the maximum number of data points to be used in the study since data could only be collected in cloud-free or broken cloud conditions, which will be further discussed in Sect. 3.2. Further constraining these distance and time conditions would eliminate more data points with negligible improvement on the statistics as shown




by Figs. S1 and S2 in the supplement. Due to missing data in the HSRL-2 data set and the removal of outliers based on collocation constraints, a total of 577 data points is available for comparison between the dropsondes and the HSRL-2 (Fig.
290    2).

An example flight, Research Flight 29 on 28 August 2020, showcases the collocation method made possible by ACTIVATE's flight strategy (Fig. 3). This flight is ideal to use for this purpose due to an above average number of dropsondes launched and its planned coordination with CALIPSO.

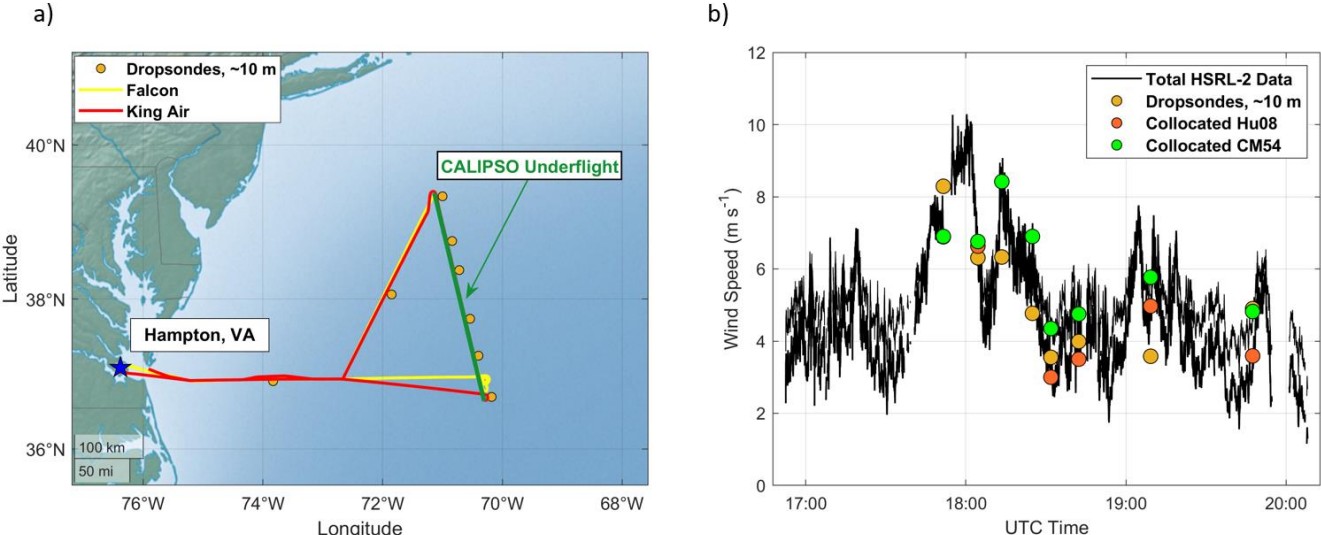

**Figure 3: a) Flight map of the King Air and Falcon from Research Flight 29 on 28 August 2020, along with dropsonde locations at the ~10 m neutral stability height, where the green line indicates the flight leg when the aircraft were spatiotemporally coordinated with the CALIPSO satellite. b) Time series of wind speed data from HSRL-2 and dropsondes for the same flight, where lines signify total HSRL-2 wind speed data and circles indicate collocated wind speed data points used in analysis. A few collocated Hu08 and CM54 wind speed data points are on top of each other owing to similar values.**

The representative flight on 28 August was a cloud-free day and consisted of the King Air and Falcon flying in the standard flight pattern termed a "statistical survey", comprised of repeated stair-stepping legs by the Falcon to probe areas within and just above the MABL (Sorooshian et al., 2023; Dadashazar et al., 2022). During this flight, the aircraft were coordinated with the CALIPSO satellite to allow comparison of ACTIVATE's remote sensing retrievals with satellite retrievals; these types of
coordinated underflights of CALIPSO were conducted in cloud-free conditions largely to intercompare aerosol measurements. Eight dropsondes were launched, six of which coincided with the CALIPSO overpass (Fig. 3a). Figure 3b highlights the results of applying this collocation method, showing that the eight collocated HSRL-2 wind speed data points agreed within ~1.2 m s$^{-1}$ of the dropsonde winds on average. However, the error between the HSRL-2 and dropsonde wind speeds reaches ~2 m s$^{-1}$ for two pairs of collocated points, which is higher than the reported error shown later in Sect. 3.1. Further analysis that
quantifies the variance of the HSRL-2 wind speed retrieval in the region of the dropsonde measurement is needed since the procedure outlined relies on collocating singular points. Therefore, information on potential geophysical variability and/or





variability in the retrieval itself may not be captured. Also, it is observed that Hu08 and CM54 wind speeds significantly differ for wind speeds below 7 m s$^{-1}$ due to the difference in parameterizations as outlined in Eq. 3.1, leading to Hu08 underestimating dropsonde wind speeds and CM54 overestimating dropsonde wind speeds at various times in Fig. 3b. Although the collocation

procedure potentially introduces additional variability in the HSRL-2 wind speed retrievals, Research Flight 29 highlights the benefits of ACTIVATE's joint deployment of the HSRL-2 and the dropsondes on the King Air for the collection of surface wind speed data.

## 2.5 Statistical Analysis Procedure

Scatterplots along with the correlation coefficient (r), linear regression, and least squares bisector regression are used to visually

demonstrate how well HSRL-2 wind speed data match dropsonde data and show any potential variability in the data. Since least squares bisector is not a common regression technique, a brief explanation of how it differs from linear regression is provided. In linear regression, $x$ is treated as the independent variable while $y$ is treated as the dependent variable. In other words, one observes how $y$ varies with changes to fixed $x$ values. In the least squares bisector method, both $x$ and $y$ are assumed to be dependent variables (Ricker, 1973). Therefore, the regression is performed assuming error is present in $x$, which accounts

for the variability in the dropsonde wind speeds. Histograms of wind speed deltas, which are defined as HSRL-2 wind speed minus dropsonde wind speed, are also created to more easily show the distribution and spread of the data. Mean, standard deviation ($STD$), 25$^{th}$ percentile ($Q1$), and 75$^{th}$ percentile ($Q3$) of the wind speed deltas are computed to quantify the bias and variability of the HSRL-2 wind speeds relative to the dropsonde wind speeds. The mean and STD are calculated as:

$$Mean = \sum_{i=1}^{N} \frac{x_{delta}}{N}, \tag{17}$$

$$STD = \sqrt{\frac{\sum_{i=1}^{N}(x_{delta} - \bar{x}_{delta})^2}{N}}, \tag{18}$$

where $N$ is the total number of wind speed data points, $x_{delta}$ is the HSRL-2 wind speed data set minus the dropsonde wind

speed data set, and $\bar{x}_{delta}$ is the mean data set of $x_{delta}$. The mean and STD are then used to calculate the error (mean ± STD) of the HSRL-2 wind speed product. Note that mean and bias are used interchangeably in the following discussion.

## 3 Results and Discussion

### 3.1 HSRL-2 – Dropsonde Comparisons

Observations collected from three years of ACTIVATE data allow for a comprehensive intercomparison of the HSRL-2

retrievals and dropsonde measurements of ocean surface wind speed. This extensive data set can also be utilized in different wind speed models (Hu08 and CM54, specifically) to observe the effects they have on these intercomparisons. All intercomparison results for both models are shown in Figs. 4 – 8 and Tables 1 – 2.







**Figure 4: Scatterplots with associated histograms for all HSRL-2 – dropsonde collocated wind speed data points for a) Hu08 and b) CM54 models. N represents the number of data points.**





**Figure 5: Scatterplots with associated histograms for Hu08 – dropsonde collocated wind speed data points for a) Wind Speed < 7 m s$^{-1}$, b) 7 m s$^{-1}$ ≤ Wind Speed < 13.3 m s$^{-1}$, and c) Wind Speed ≥ 13.3 m s$^{-1}$. Note that x- and y-axis ranges vary to better showcase results in individual panels.**






**Figure 6: Scatterplots with associated histograms for CM54 – dropsonde collocated wind speed data points for a) Wind Speed < 7 m s⁻¹, b) 7 m s⁻¹ ≤ Wind Speed < 13.3 m s⁻¹, and c) Wind Speed ≥ 13.3 m s⁻¹. Note that x- and y-axis ranges vary to better showcase results in individual panels.**







Figure 7: Scatterplots with associated histograms for Hu08 – dropsonde collocated wind speed data points for a) winter and b) summer deployments.







**Figure 8: Scatterplots with associated histograms for CM54 – dropsonde collocated wind speed data points for a) winter and b) summer deployments.**



**Table 1a: Summary of all Hu08 – dropsonde wind speed statistics shown in Figs. 4, 5, 7 scatterplots. The two values for slope and y-intercept refer to those for the linear and bisector regressions, in that order.**

|  | N | Slope | Y-intercept [m s$^{-1}$] | r |
|---|---|---|---|---|
| Overall | 577 | 1.04/1.17 | -0.13/-1.05 | 0.89 |
| Wind Speed < 7 m s$^{-1}$ | 292 | 0.65/0.99 | 1.10/-0.49 | 0.66 |
| 7 m s$^{-1}$ ≤ Wind Speed < 13.3 m s$^{-1}$ | 236 | 0.64/0.85 | 3.80/1.87 | 0.75 |
| Wind Speed ≥ 13.3 m s$^{-1}$ | 49 | 0.04/0.94 | 14.57/3.01 | 0.06 |
| Winter | 236 | 0.95/1.08 | 1.03/-0.08 | 0.88 |
| Summer | 341 | 1.08/1.24 | -0.69/-1.68 | 0.87 |

**Table 1b: Summary of all Hu08 – dropsonde wind speed statistics shown in Figs. 4, 5, 7 histograms.**

|  | N | Mean [m s$^{-1}$] | STD [m s$^{-1}$] | 25$^{th}$ perc. [m s$^{-1}$] | 75$^{th}$ perc. [m s$^{-1}$] | Mean Error [m s$^{-1}$] |
|---|---|---|---|---|---|---|
| Overall | 577 | 0.15 | 1.80 | -0.86 | 1.13 | 0.15 ± 1.80 |
| Wind Speed < 7 m s$^{-1}$ | 292 | -0.54 | 1.34 | -1.27 | 0.22 | -0.54 ± 1.34 |
| 7 m s$^{-1}$ ≤ Wind Speed < 13.3 m s$^{-1}$ | 236 | 0.56 | 1.49 | -0.31 | 1.46 | 0.56 ± 1.49 |
| Wind Speed ≥ 13.3 m s$^{-1}$ | 49 | 2.24 | 2.97 | 0.24 | 3.38 | 2.24 ± 2.97 |
| Winter | 236 | 0.63 | 2.07 | -0.49 | 1.72 | 0.63 ± 2.07 |
| Summer | 341 | -0.18 | 1.52 | -1.01 | 0.71 | -0.18 ± 1.52 |


**Table 2a: Summary of all CM54 – dropsonde wind speed statistics shown in Figs. 4, 6, 8 scatterplots. The two values for slope and y-intercept refer to those for the linear and bisector regressions in that order.**

|  | N | Slope | Y-intercept [m s$^{-1}$] | r |
|---|---|---|---|---|
| Overall | 577 | 0.91/1.03 | 1.28/0.37 | 0.88 |
| Wind Speed < 7 m s$^{-1}$ | 292 | 0.49/0.79 | 2.76/1.32 | 0.63 |
| 7 m s$^{-1}$ ≤ Wind Speed < 13.3 m s$^{-1}$ | 236 | 0.64/0.85 | 3.80/1.87 | 0.75 |
| Wind Speed ≥ 13.3 m s$^{-1}$ | 49 | 0.01/0.99 | 14.77/2.39 | 0.01 |
| Winter | 236 | 0.86/0.99 | 2.10/0.98 | 0.87 |
| Summer | 341 | 0.88/1.04 | 1.14/0.16 | 0.85 |

**Table 2b: Summary of all CM54 – dropsonde wind speed statistics shown in Figs. 4, 6, 8 histograms.**





|  | N | Mean [m s$^{-1}$] | STD [m s$^{-1}$] | 25$^{th}$ perc. [m s$^{-1}$] | 75$^{th}$ perc. [m s$^{-1}$] | Mean Error [m s$^{-1}$] |
|---|---|---|---|---|---|---|
| Overall | 577 | 0.62 | 1.70 | -0.28 | 1.43 | 0.62 ± 1.70 |
| Wind Speed < 7 m s$^{-1}$ | 292 | 0.35 | 1.30 | -0.28 | 1.14 | 0.35 ± 1.30 |
| 7 m s$^{-1}$ ≤ Wind Speed < 13.3 m s$^{-1}$ | 236 | 0.56 | 1.49 | -0.31 | 1.46 | 0.56 ± 1.49 |
| Wind Speed ≥ 13.3 m s$^{-1}$ | 49 | 2.20 | 2.87 | 0.16 | 3.55 | 2.20 ± 2.87 |
| Winter | 236 | 0.91 | 2.04 | -0.23 | 1.92 | 0.91 ± 2.04 |
| Summer | 341 | 0.42 | 1.38 | -0.31 | 1.17 | 0.42 ± 1.38 |

Strong correlations are seen between dropsonde winds and both Hu08 and CM54 as seen in Fig. 4, with the correlation coefficients being 0.89/0.88 (Hu08/CM54). Biases are 0.15/0.62 m s$^{-1}$ while the STDs are 1.81 m s$^{-1}$/1.80 m s$^{-1}$. Regression y-intercepts (linear/bisector) are also similar between the two models (-0.13/-1.05 m s$^{-1}$ for Hu08 and 1.28/0.37 m s$^{-1}$ for CM54), although Hu08 has slightly better performance in the linear fit while CM54 seems to perform better in the bisector fit. As seen by mean and STD values shown in Fig. 4, the error of Hu08 wind speeds is 0.15 m s$^{-1}$ ± 1.80 m s$^{-1}$ while the error of CM54

wind speeds is 0.62 m s$^{-1}$ ± 1.70 m s$^{-1}$. These results show that both Hu08 and CM54 wind speeds are biased slightly high compared to the dropsonde wind speeds, which is probably due to the influence of high wind speed outliers seen in both the scatterplots and histograms.

To further investigate the source of these bias and variability trends, the data are divided into low (Wind Speed < 7 m s$^{-1}$),

intermediate (7 m s$^{-1}$ ≤ Wind Speed < 13.3 m s$^{-1}$), and high (Wind Speed ≥ 13.3 m s$^{-1}$) wind speed regimes as shown in Figs. 5 and 6. These regimes correspond with those that Hu08 delineates in Eqs. 3.1 – 3.3. In the low wind regime, Hu08 is biased low at -0.54 m s$^{-1}$ while CM54 winds are biased high at 0.35 m s$^{-1}$. Also, the low wind speed regime intercomparisons have the lowest variability compared to those of the intermediate and high wind speed regimes in terms of STD and y-intercept. Based on these observations, the low bias and variability of Hu08 winds below 7 m s$^{-1}$ drive down the bias and variability in

the overall data comparisons. At the same time, both Hu08 and CM54 wind speeds in the high wind speed regime have relatively poor agreement with dropsonde values and increase the bias and variability observed in the intercomparisons. However, most of the data are contained within the low wind speed regime, which mostly offset errors observed in the high wind speed regime.

The wind speed data are also divided into winter and summer deployments (dates provided in Sect. 2.2) as shown in Figs. 7 and 8 to determine if the wind speed intercomparisons have dependence on seasonal factors. Overall, summer intercomparisons perform better than winter ones based on summer's lower bias (-0.18/0.42 m s$^{-1}$) and STD (1.52/1.38 m s$^{-1}$) values compared to winter's bias (0.63/0.91 m s$^{-1}$) and STD (2.07/2.04 m s$^{-1}$) values. Therefore, HSRL-2 wind speed error is higher for winter comparisons (0.63 m s$^{-1}$ ± 2.07 m s$^{-1}$/0.91 m s$^{-1}$ ± 2.04 m s$^{-1}$) than for summer comparisons (-0.18 m s$^{-1}$ ± 1.52 m s$^{-1}$/0.42 m s$^{-}$





[1] $\pm$ 1.38 m s$^{-1}$). These results make sense because of the more turbulent and windy conditions in winter (i.e., relatively more wind speeds above 13.3 m s$^{-1}$), as discussed in Sect. 2.2, in contrast to summer months that coincide with a higher fraction of data representing wind speeds below 7 m s$^{-1}$. Despite the winter HSRL-2 wind speed retrievals having higher error than the summer ones, HSRL-2 wind speeds in the winter are still within ~0.9 m s$^{-1}$ of the dropsonde wind speeds.

### 3.2 Case Studies

Wind speed data from two ACTIVATE research flights are analyzed (Research Flight 14 on 1 March 2020 and Research Flight 100 on 11 January 2022) to 1) demonstrate the special ability of the HSRL-2 to provide wind speed profiles that show the spatial variability of wind speed over time, which are beneficial to observe phenomena such as cloud evolution and sea-surface temperature dynamics and 2) identify potential drawbacks of using this retrieval method during days with high cloud fraction conditions. First, the 1 March 2020 case study is shown in Fig. 9 to address how HSRL-2 is beneficial for studying cloud and

sea surface temperature dynamics.





**Figure 9: a) Flight map of Falcon (yellow line), King Air (red line), and dropsondes (dark yellow circles) overlaid onto Geostationary**
**Operational Environmental Satellite (GOES-16) cloud imagery for Research Flight 14 on 1 March 2020. Blue stars represent time**
**stamps where the aircraft cross over from cloud-free to cloudy areas. b) Flight map overlaid onto map of MERRA-2 mean sea-**
**surface temperature data (GMAO, 2015). Purple stars represent time stamps where the aircraft cross over sharp sea surface**
**temperature changes associated with the Gulf Stream. c) Time series of wind speed data from HSRL-2 and dropsondes for the same**
**flight, where lines signify total HSRL-2 wind speed data and circles indicate collocated wind speed data points used in analysis.**
**Vertical lines represent time stamps of interest as indicated in a) and b). Collocated Hu08 and CM54 wind speed data points are on**
**top of each other for the first pair of points in order of time.**

Included are the flight tracks overlaid on GOES imagery and sea surface temperature data from Modern-Era Retrospective

Analysis for Research and Applications, Version 2 (MERRA-2) (Gelaro et al., 2017) (Fig. 9a-b). This flight and the associated

morning flight on that same day have been the subject of several studies owing to its coincidence with cold air outbreak

conditions (see cloud streets in Fig. 9a) and a flight strategy that allowed for detailed characterization of the evolving aerosol-

cloud system as a function of distance offshore (Chen et al., 2022; Li et al., 2022; Tornow et al., 2022; Sorooshian et al., 2023).





There was a significant sea surface temperature gradient this day (Fig. 9b), which is common along the Gulf Stream border. The morning flight focused on a location with very detailed characterization including stacked level flight legs (i.e., termed a "wall") with the Falcon flying below, in, and above clouds with the King Air flying aloft to further characterize the same
region. The afternoon flight consisted of both aircraft flying back to that same location, adjusting the sampling strategy to fly along the boundary layer wind direction in a quasi-Lagrangian fashion to keep studying the evolution of the air mass characterized in the morning. This case shows how the HSRL-2 samples in a broken-cloud field and provides a profile of how surface wind speeds spatially vary with time.

A special benefit of the HSRL-2 surface wind speed time series is the significant spatial gradient in surface wind speeds captured that otherwise would not be available solely with the dropsondes. The AVAPS dropsonde measurements provide data only at single points and cannot provide the spatial extent that the derived HSRL-2 surface wind speeds can contribute. Although the sharpest changes in the cloud field, sea surface temperature, and surface wind speed (Fig. 9) do not match exactly in space, surface wind speeds play an important role in the interplay between clouds, sea surface temperature, and MABL
dynamics. This flight consisted of significantly higher wind speeds farther offshore, consistent with strong heat fluxes and postfrontal cloud development. Boundary layer wind speed and direction shear are key parameters relevant to understanding cloud formation, morphology, and life cycle during cold air outbreaks (e.g., Chen et al., 2022). In Fig. 9c, the two collocated wind data point pairs agreed within (in order of time) 1.80 m s$^{-1}$ and 3.32 m s$^{-1}$; these agreements are less than what was observed in Fig. 3c. The agreement between HSRL-2 and dropsonde surface wind speeds is better at lower wind speeds as
mentioned in Sect. 3.1, consistent with how the agreement for the first of the two pairs of collocated points in Fig. 3c was better as it was at lower wind speed.

It may be difficult to use HSRL-2 wind speed retrievals during days with high cloud fraction conditions. As mentioned in Sect. 3.1, there is poor agreement for wind speeds in the high wind regime relative to wind speeds in the low and intermediate





regimes. One source of these outliers comes from Research Flight 100 conducted on 11 January 2022 with the outliers being

highlighted in Fig. 10.

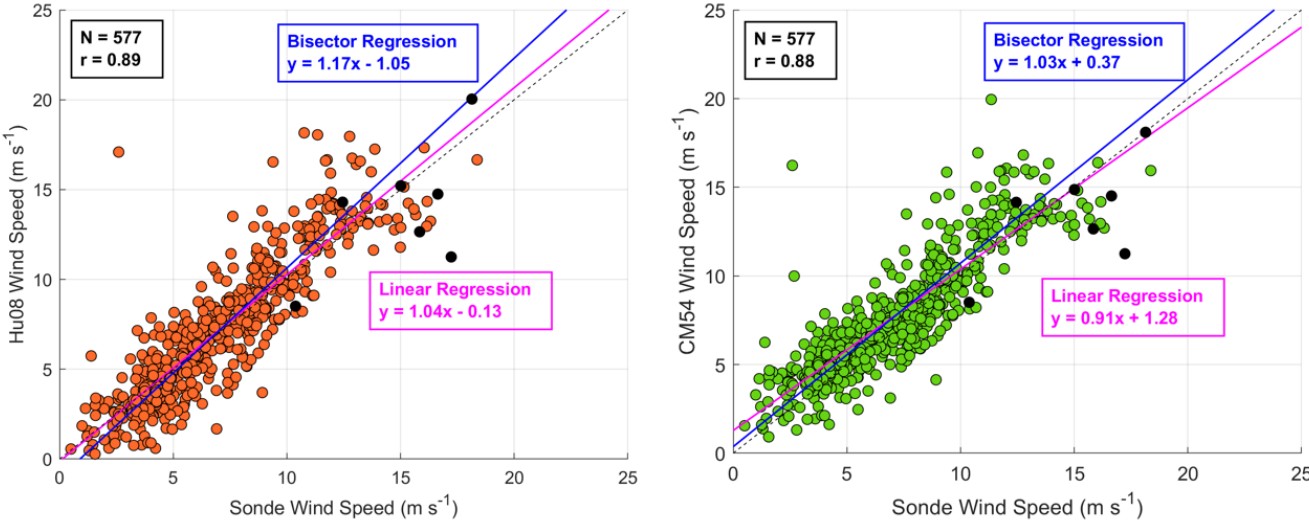

**Figure 10: Hu08 and CM54 scatterplots from Fig. 4. with wind speed data points from 11 January 2022 highlighted using black circles.**

Figure 10 shows that this research flight contributes to some of the largest outliers seen in Fig. 4. This day was deemed as an

excellent cold air outbreak day due to significant temperature gradients at the air-sea interface, leading to a deep boundary

layer, which strongly impacted the evolution of clouds on this day. Cloud coverage was significant (Fig. 11) along with visible

steam fog and numerous whitecaps at the ocean surface. Precipitation and ice were also present, leading to icing on the aircraft

instruments.




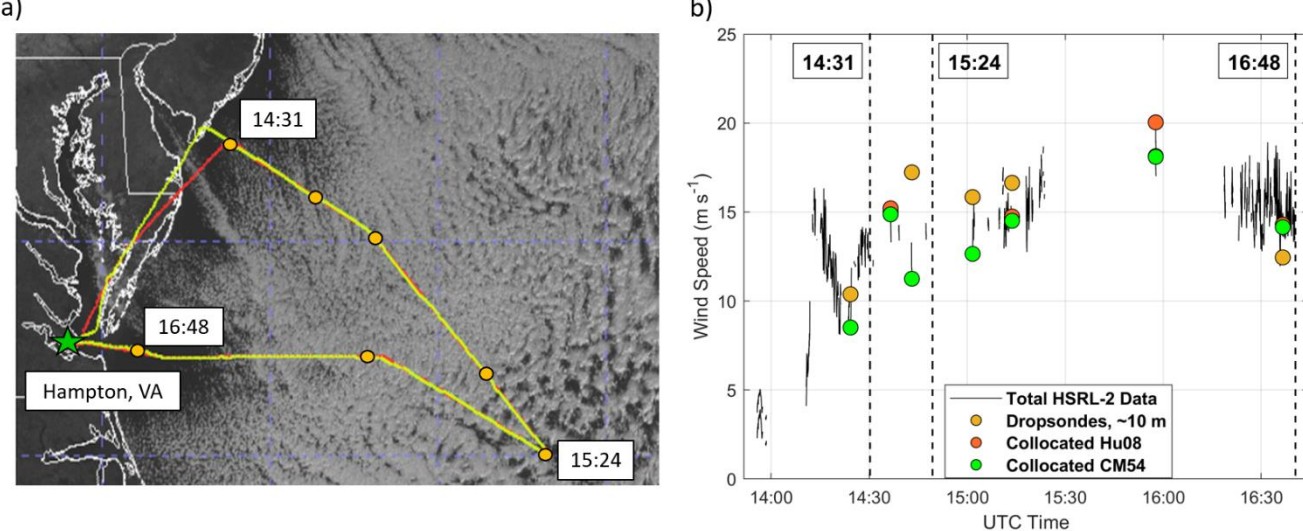

**Figure 11: a) Flight map of Falcon (yellow line), King Air (red line), and dropsondes (dark yellow circles) overlaid onto Geostationary Operational Environmental Satellite (GOES-16) cloud imagery for Research Flight 100 on 11 January 2022. Time stamps at 14:31 and 16:48 represent areas where HSRL-2 was able to retrieve wind speed data while the 15:24 time stamp shows where HSRL-2 had difficulty with wind speed retrievals. b) Time series of wind speed data from HSRL-2 and dropsondes for the same flight, where lines signify total HSRL-2 wind speed data and circles indicate collocated wind speed data points used in analysis. Vertical lines represent time stamps of interest as indicated in a). Some collocated Hu08 and CM54 wind speed points are on top of each other owing to similar values.**

As seen in Fig. 11b, most of the HSRL-2 wind speed data were missing during this flight due to quality control standards in the cloud screening of the data. If a wind speed retrieval is taken in an area with a high cloud fraction, which dominated most of the flight day (Fig. 11a), the retrieval is deemed a missing value. Therefore, it was more difficult to collocate dropsonde wind speeds with the Hu08 and CM54 wind speeds for a majority of the flight. The largest disagreement between HSRL-2 and dropsonde wind speeds is seen at 15:24 (Fig. 11b), occurring at an area with high cloud fraction and in turn had very limited HSRL-2 wind speed data that fit within the collocation criteria. The HSRL-2 had its most successful retrievals at 14:31 and 16:48 (Fig. 11a), which coincided with areas of less cloud cover. Although the HSRL-2 wind speed retrievals showed great success overall as seen in Sect. 3.1, this product is best used on days with cloud-free or broken cloud conditions.

## 4. Conclusions

This study introduces the High Spectral Resolution Lidar – generation 2 (HSRL-2) ocean surface wind speed retrieval method and demonstrates its use and accuracy using data collected during the NASA ACTIVATE field campaign from 2020 to 2022. Specifically, the capabilities of the HSRL-2 onboard the King Air are assessed by spatiotemporally collocating and comparing its data to NCAR AVAPS dropsonde measurements. This work presents the HSRL-2 retrieval algorithm using the Hu (Hu08) wind speed – wave-slope variance model with these results being compared with wind speeds using the well-known Cox-Munk (CM54) model. HSRL-2 wind speeds are strongly correlated with dropsonde wind speeds (r: 0.89/0.88) (Hu08/CM54)





and these retrievals have an overall error of 0.15 m s$^{-1}$ ± 1.80 m s$^{-1}$ for Hu08 and 0.62 m s$^{-1}$ ± 1.70 m s$^{-1}$ for CM54. Results are also shown for wind speeds divided into low (Wind Speed < 7 m s$^{-1}$), intermediate (7 m s$^{-1}$ ≤ Wind Speed < 13.3 m s$^{-1}$), and high (Wind Speed ≥ 13.3 m s$^{-1}$) categories. It is observed that high HSRL-2 winds have higher bias and variability compared to low HSRL-2 wind speeds. However, high HSRL-2 wind speeds are still within 2.20 m s$^{-1}$ of the dropsonde wind speeds. Seasonal intercomparisons are also performed by categorizing the wind speed data into winter and summer deployment

periods. The results of those comparisons indicate that HSRL-2 wind speeds in the winter have higher error (0.63 m s$^{-1}$ ± 2.07 m s$^{-1}$/0.91 m s$^{-1}$ ± 2.04 m s$^{-1}$) than in the summer (-0.18 m s$^{-1}$ ± 1.52 m s$^{-1}$/0.42 m s$^{-1}$ ± 1.38 m s$^{-1}$), due in part to the winter having a higher frequency of wind speeds in the high wind speed regime whereas the summer contains more wind speeds in the low regime. Although the Hu08 results show a lower error, this study does not definitively conclude that either the Hu08 or CM54 model is better based on the statistics shown in Sect. 3.1.


This novel retrieval method offers a new path forward in airborne field work for the acquisition of ocean surface wind speed data at a high time resolution (10 s), as demonstrated with two case study flights (Research Flight 14 on 1 March 2020 and Research Flight 29 on 28 August 2020). Having such data can benefit several scientific applications related to air-sea interactions such as estimating heat fluxes, gas exchange, sea salt emissions and aerosol transport, and cloud life cycle.

However, another case study flight (Research Flight 100 on 11 January 2022) shows that the HSRL-2 wind speed retrievals are limited on days with high cloud fraction where the lidar signal is highly attenuated at the surface.

Forthcoming work will continue assessments of wind speed measurements during ACTIVATE by comparing dropsonde data to in situ measurements taken by the Turbulent Air Motion Measurement System (TAMMS) onboard the Falcon aircraft at its

various altitude legs (between 120 m and 5 km) (Thornhill et al., 2003). Comparisons will also be performed between TAMMS wind speeds and wind speed data from reanalysis models such as the Modern-Era Retrospective analysis for Research and Applications, version 2 (MERRA-2). Intercomparisons with MERRA-2 will be particularly important because this reanalysis model was used in NASA's Cloud-Aerosol Lidar and Pathfinder Satellite Observation (CALIPSO) retrievals of AOD using the surface scattering and modelled wind speed from MERRA-2. Additional work is also warranted to assess the wind speed

retrievals performed by ACTIVATE's other remote sensor, the Research Scanning Polarimeter (RSP), to fully demonstrate ACTIVATE's remote sensing capabilities. By evaluating these airborne wind speed measurement techniques, it is hoped that they can be reliably used in future field campaigns to obtain accurate, high resolution wind speed data to further enhance science investigations.

**Data Availability**

ACTIVATE airborne data are available through https://asdc.larc.nasa.gov/project/ACTIVATE (ACTIVATE Science Team, 2020). MERRA-2 mean sea surface temperature data are taken from the 2d, 1-Hourly, Time-Averaged, Single-Level,



Assimilation, Surface Flux Diagnostics V5.12.4 (M2T1NXFLX) product found at https://disc.gsfc.nasa.gov/datasets/M2T1NXFLX_5.12.4/summary (doi.org/10.5067/7MCPBJ41Y0K6). GOES-16 data are from https://asdc.larc.nasa.gov/ACTIVATE/ACTIVATE-Satellite_1 (doi: 520 10.5067/ASDC/SUBORBITAL/ACTIVATE-Satellite_1).

**Author Contribution**

SD performed all analyses with input from all co-authors. SD, JH, RF, CH, JR, and AS prepared manuscript with all co-authors involved in review/editing. TS, DH, SS, and CR conducted flight scientist duties on the King Air and helped with preparation and deployment of dropsondes. JH, RF, MF, CH, BC, DH, SS, TS, and EC were responsible for the HSRL-2 instrumentation 525 and KT, CR, and HV were responsible for the NCAR AVAPS dropsonde instrumentation collected and subsequent archival of wind speed data sets needed to conduct this analysis. JH, JR, YH, RF, CH, and BC all contributed with formulation of HSRL-2 retrieval algorithm.

**Competing Interests**

The authors declare that they have no conflict of interest.

**Disclaimer**

Publisher's note: Copernicus Publications remains neutral with regard to jurisdictional claims in published maps and institutional affiliations.

**Acknowledgements**

The work was funded by ACTIVATE, a NASA Earth Venture Suborbital-3 (EVS-3) investigation funded by NASA's Earth 535 Science Division and managed through the Earth System Science Pathfinder Program Office. We thank pilots and aircraft maintenance personnel of NASA Langley Research Services Directorate for successfully conducting ACTIVATE flights. The MERRA-2 1-Hourly, Time-Averaged, Single-Level, Assimilation, Surface Flux Diagnostics V5.12.4 (M2T1NXFLX) data used in this effort were acquired as part of the activities of NASA's Science Mission Directorate and are archived and distributed by the Goddard Earth Sciences (GES) Data and Information Services Center (DISC).

**Financial support**

University of Arizona investigators were funded by NASA grant no. 80NSSC19K0442.



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
