# Peer review of "High Spectral Resolution Lidar – generation 2 (HSRL-2) Retrievals of Ocean Surface Wind Speed: Methodology and Evaluation"

_EGUsphere, 2023_

## Referee Comment (RC2)

**Review of egusphere-2023-1943: "HSRL-2 Retrievals of Ocean Surface Wind Speeds" by Dmitrovic et al.**

In this work the authors describe an algorithm for deriving ocean surface wind speed estimates from measurements of ocean surface backscatter acquired by the NASA-Langley high spectral resolution lidar (HSRL). The NASA HSRL can accurately characterize the signal attenuation above the ocean surface and reliably partition the surface signal pulse into pure surface and ocean subsurface components, and hence can deliver high quality measurements of surface integrated attenuated backscatter ($\beta_{surf}$). The accuracy of the wind speed retrieval thus depends on the equation relating $\beta_{surf}$ to wave slope variances and the fidelity of the model used to convert wave slope variances to wind speeds. The authors derive wind speeds using two different models – the classic Cox-Munk (1954) and a lidar-specific model developed by Hu et al. (2008) – and compare these results to near-simultaneous dropsonde measurements of wind speeds. The paper is well organized and well written and its subject matter is entirely appropriate for Atmospheric Measurements Techniques. There are, however, several issues that should be addressed prior to publication.

My primary concern is that the authors' equation (2) fails to acknowledge the possible presence of whitecaps and/or sea foam. While the authors cite Josset et al., 2010b as their source for equation (2), that work explicitly includes reflection from the whitecap fraction within any footprint; see section 2.2 and equations (2) and (21) therein. Furthermore, comparing the upper and lower panels of figure (2) in Hu et al., 2008 suggests that omitting whitecap contributions could have a significant impact on the results derived in this paper. On the other hand, Lancaster et al. (2005) suggest that the "contribution of whitecaps to the nadir lidar measurements is seen in Figure 2 to be negligible".

I note that the authors' equation (2) also omits the atmospheric two-way transmittance term give in equation (21) in Josset et al., 2010b. Given the discussion in the introduction about calibration transfer and the assertion on line 116, this omission seems a bit surprising.

When considering the authors' wind speed difference statistics, I kept wondering about wind speed variations over time at a fixed point. HSRL biases relative to the dropsondes are given as either $0.15 \pm 1.80$ m/s or $0.62 \pm 1.70$ m/s, depending on the model used. But the temporal offset between matched HSRL and dropsonde wind speed estimates can be as large as 15 minutes. How do these bias magnitudes compare to the natural variations in wind speed that would be measured at a fixed point over a 15-minute time interval? Perhaps wind speed variability information is readily available from the NOAA's National Data Buoy Center? A box and whisker plot showing wind speed differences as a function of temporal offset between the two data sets might also shed some light on this issue.

On lines 51–52, immediately after describing the rudiments of Hu et al., 2008 derivation, the authors introduce equation (1) by say, "The wind speed (U) was then approximated from the wave-slope variance ($\sigma^2$) through this linear relationship". What I was expecting to see were the Hu equations subsequently given as equations (3.1) through (3.3). Instead, equation (1) is Cox-Munk. This section of the text (lines 44–52) should be rewritten to clearly distinguish between the original Cox-Munk equation and the subsequent CALIPSO derivation by Hu.

From a quick glance at the studies cited on line 54, I do not find any support for the assertion that "CALIPSO retrievals of surface wind speeds have been used in many studies". Josset et al., 2010a used AMSR winds to investigate "the normalized scattering cross section" of the CALIPSO lidar and the CloudSat radar. Kiliyanpilakkil and Meskhidze used AMSR winds and aerosol optical properties derived from CALIPSO. Nair & Rajeev used QuickScat winds and CALIPSO cloud heights. Sun et al. uses "numerical weather prediction wind vector assimilated with observed wind component" obtained from ALADIN.

Figure 1 and its supporting description are all very nicely done. I commend the authors for their clear and informative presentation of this material.

Please provide an overview of the primary sources of uncertainty associated with calculating $\beta_{surf}$ using equation (15). How is $\Delta\beta_{surf}$ estimated? In practice, is there some maximum $\Delta\beta_{surf} / \beta_{surf}$ above which the retrieval is deemed too unreliable for subsequent wind speed estimation?

Figures 4–7: ordinary least squares problems can be extremely sensitive to large outliers. Did the authors consider applying an outlier rejection scheme (e.g., Tukey fencing) before computing the regression lines shown in these figures?

Lines 470–477: I am totally bewildered by the authors' data screening criteria; i.e., "if a wind speed retrieval is taken in an area with a high cloud fraction [...] the retrieval is deemed a missing value". Please explain what is meant by "cloud fraction" in this context. Is this vertical cloud fraction within an individual HSRL profile? Perhaps naively, I would think that (a) wind speed retrievals would be possible any time the ocean surface was reliably detected and (b) a much better QA metric could be derived from the quality of the surface backscatter signal.

Minor Remarks

The formatting of equation (1) is ambiguous. Decimal notation would be much, much better I think (e.g., 0.003 instead of 3.0E – 3).

Line 105: what value did the authors use for the Fresnel coefficient? Note: Hu et al., 2009 use 0.0209, Josset et al., 2010a use 0.0213, and Venkata and Reagan 2015 use 0.0205.

Line 105: what is the typical off-nadir angle for the HSRL measurements? Should readers assume nadir pointing, so that $\theta = 0$?

Line 110: "Eq. 3.3 is  identical to the log-linear relationship proposed by Wu (1990)."

Line 111: change "to be" to "being"

Line 192: practically speaking, is there some maximum AOD above which surface wind speeds are not considered reliable? Or are there perhaps some meteorological conditions in which the method is not applicable (e.g., exceptionally dense surface-hugging fogs)?

Figure 3: use different line colors and/or line types to plot the two different sets of HSRL wind speed retrievals.

Lines 299–300: In the figure caption, the authors say, "A few collocated Hu08 and CM54 wind speed data points are on top of each other owing to similar values." They could (and should) eliminate any ambiguity by specifying UTC for these pairs of points.

Lines 319–325: I would have appreciated a bit more detail here. The authors' description does not provide sufficient information to distinguish the bisector method from other 'errors in variables' techniques (e.g., Deming regression and orthogonal distance regression). Is the bisector method especially effective for problems of this sort? Or will any errors in variables method do equally well?

**One Reviewer's Opinion**

A scatter plot of dropsonde wind speeds (U) versus matching values of $\sigma^2$ (derived using equation (3) and $\beta_{surf}$ computed using equation (15)) would have made this paper enormously more interesting. Both Cox-Munk and Hu et al., 2008 are approximations of the true relationship between wave slope variance and surface wind speeds. The collocated measurements reported in this manuscript offer a superb opportunity to evaluate the relative merits of both models. Perhaps this tantalizing topic can be briefly explored in an appendix included in a revision to the current manuscript.

**References**

Cox and Munk, 1954: Measurement of the Roughness of the Sea Surface from Photographs of the Sun's Glitter, *J. Opt. Soc. Am.*, **44**, 838–850, https://doi.org/10.1364/JOSA.44.000838.

Hu et al., 2008: Sea surface wind speed estimation from space-based lidar measurements, *Atmos. Chem. Phys.*, **8**, 3593–3601, https://doi.org/10.5194/acp-8-3593-2008.

Josset et al., 2010a: Multi-Instrument Calibration Method Based on a Multiwavelength Ocean Surface Model, *IEEE Geosci. Remote Sens. Lett.*, **7**, 195–199, https://doi.org/10.1109/LGRS.2009.2030906.

Josset et al., 2010b: Lidar equation for ocean surface and subsurface, *Opt. Express*, **18**, 20862–20875, https://doi.org/10.1364/OE.18.020862.

Lancaster et al., 2005: Laser pulse reflectance of the ocean surface from the GLAS satellite lidar, *Geophys. Res. Lett.*, **32**, L22S10, https://doi.org/10.1029/2005GL023732.

Venkata and Reagan, 2016: Aerosol Retrievals from CALIPSO Lidar Ocean Surface Returns, *Remote Sens.*, **8**, 1006, https://doi.org/10.3390/rs8121006.

---

## Author Comment (AC1)

We thank the reviewer for their thoughtful suggestions and constructive criticism that have helped us improve our manuscript. Below we provide responses to reviewer concerns and suggestions in blue font.

**Original Submission**

**HSRL-2 Retrievals of Ocean Surface Wind Speeds**

**1.1. Recommendation**

**Major revision**

1. **Comments to Author:**

**Overall opinion:** This paper demonstrates the HSRL-2 retrieval of ocean surface wind speeds based on the HSRL-2 measurement principle relying on the wave-slope variance determination and dedicated campaign of comparison with the airborne and dropsonde data. The topic you cover is a key for understanding the response of ocean surface reflectivity to changing conditions of ocean and elucidates previously unknown aspects on ocean surface wind speed using HSRL. You also have nearly perfect instrumentation to address this issue. However, this study is not ready to be accepted due to the following setbacks: the plain title that carries very little information about the exact, less broad scope or the actual content of the study, the lack of quantitative information in the abstract (you are not convincing by simply reporting four numerical arguments as your main proofs), poorly structured methodology with significant gaps (did you specify the angular specifications of HSRL-2 system for instance?) within and most critically, unconvincing "results" section due to poor structural choices and omitted numerical arguments in some cases when you speak about correlation. Please revise the manuscript using the comments below.

**2.1. Comments:**

**Title:** The title is overly-general. It sounds like it's a white paper on HSLR retrieval on ocean surface winds. First, I am not sure whether it is a good idea to use unexplained acronyms even in the journals like AMT oriented on a very specific niche of experts. Moreover, what have you exactly done with HSRL-2 retrievals, introduced them? Evaluated? For how long period of time, etc? Your title does not reflect this idea and is therefore not a good title choice.

Thank you for your feedback on the title. We've changed it to "High Spectral Resolution Lidar – generation 2 (HSRL-2) Retrievals of Surface Wind Speeds: Methodology and Evaluation" to be more specific on the paper's aims as you suggested.

**Abstract:** The abstract is not convincing in the present form because of the following issues:

- No Justification. Why it is important to understand, for instance, sea surface reflectance or surface wind speeds over ocean using HSRL? Which research gap you will close by bringing new knowledge on this topic? The statement, explaining this aspect should start your abstract from my point of view if you think about general readers.

  We appreciate you bringing up this important point. We added this justification to the beginning of the abstract.

  Added: "Ocean surface wind speed (i.e., wind speed 10 m above sea level) is a critical parameter used by atmospheric models to estimate the state of the marine atmospheric boundary layer (MABL). Accurate surface wind speed measurements in diverse locations are required to improve characterization of MABL dynamics and assess how models simulate large-scale phenomena related to climate change and global weather patterns. To provide these measurements, this study introduces and evaluates a new surface wind speed data product from NASA Langley Research Center's High Spectral Resolution Lidar – generation 2 (HSRL-2) using data collected as part of NASA's Aerosol Cloud meTeorology Interactions oVer the western ATlantic Experiment (ACTIVATE) mission."

- Poor structure: Unclear role of two research flights that were introduced after you reported some results in numerical form. It's uncommon structural decision for abstract of a peer-reviewed study to juggle between the results and methodological descriptions. Both introduction and main text leaves the feeling that two sections of the scientific analysis are poorly connected to each other, please address this aspect.

  We agree that the way the two research flights were introduced was confusing. The goal of these case studies is to show how the HSRL-2 can continuously sample surface wind speeds on a given flight day, even on days with clouds present. We present this information more succinctly as follows:

  Revised: "Also, the high horizontal spatial resolution of the HSRL-2 retrievals (0.5 s or ~75 m along track) allows the instrument to probe the fine-scale spatial variability of surface wind speeds over time along the flight track and breaks in broken cloud fields."

- Scientific value: You reported four numbers numerically in an abstract of the study, called "HSRL-2 retrievals". A simple question here: can we actually make any conclusions about HSRL-2 retrievals based on four numbers? You may reconsider the structure of your manuscript by minimizing methodological information in brief way and by nailing down your numerical arguments about efficiency of HSRL-2 retrieval.

  Thank you for this advice. We agree that our numerical arguments should clearly show how the HSRL-2 retrievals perform. Therefore, we include the results of the retrieval assessment when the data is separated by 1) wind speed regime and 2) season. The results of the model comparison are also included to show how the Hu et al. (2008) model performs better for surface wind speeds below 7 m s$^{-1}$.

Revised: "These comparisons show correlations of 0.89, slopes of 1.04 and 1.17, and y-intercepts of -0.13 m s$^{-1}$ and -1.05 m s$^{-1}$ for linear and bisector regressions, respectively and the overall accuracy is calculated to be 0.15 m s$^{-1}$ ± 1.80 m s$^{-1}$. It is also shown that the dropsonde surface wind speed data most closely follows the HSRL-2 distribution of wave-slope variance using the distribution proposed by Hu than the ones proposed by Cox-Munk and Wu distributions for surface wind speeds below 7 m s$^{-1}$, with this category comprising most of the ACTIVATE data set. The retrievals are then evaluated separately for surface wind speeds below 7 m s$^{-1}$ and between 7 m s$^{-1}$ and 13.3 m s$^{-1}$ and show that the HSRL-2 retrieves surface wind speeds with a bias of ~0.5 m s$^{-1}$ and an error of ~1.5 m s$^{-1}$, a finding not apparent in the cumulative comparisons. Also, it is shown that the HSRL-2 retrievals are more accurate in the summer (-0.18 m s$^{-1}$ ± 1.52 m s$^{-1}$) than winter (0.63 m s$^{-1}$ ± 2.07 m s$^{-1}$), but the HSRL-2 is still able to make numerous, (N = 236) accurate retrievals in the winter."

- You implied the wind speed over ocean behind the term "wind speed" but not articulated in sufficiently for a general reader by quickly resorting to the term without mentioning "ocean". There is no value in wave-slope parametrizations over land, right? Please either emphasize once for the entire manuscript that you mean wind speed over ocean or always stick to this term, please.

  Thank you, we agree that wind speed needs to be better articulated. We now introduce the term "ocean surface wind speed" as wind speeds above 10 m above sea level, then transition (with notice to the reader) into "surface wind speed" throughout the abstract and the entire paper.

**Introduction:** Several non-critical issues here:

- You spoke about CALIPSO retrievals of surface wind speeds using Cox-Munk principle but forgot about one of the seminal works on this topic (Josset et al., 2008 about synergy of CALIPSO and cloud radar data to retrieve AOD from this relationship). So, your speculations about correction of ocean signal by available AOD information from CALIPSO should definitely point out to this study and mention please that this idea has been already proposed and utilized by researchers.

  Thank you for bringing up. We now include Josset et al. (2008) when explaining the CALIPSO retrievals.

  Revised: "If coincident aerosol optical depth (AOD) data are available (e.g., from MODIS in the case of CALIPSO detailed in Josset (2008)) then they may be used to estimate the intervening attenuation and transfer the calibration."

- CM54 as a term is actually worse choice than just plainly referring to the Cox-Munk principle as 'Cox-Munk' or 'Cox and Munk parametrization' from the reader point of view. You save very little space by introducing an unnecessary acronym that has to be searched in the manuscript by a reader. Moreover, Cox-Munk principle is quote known in this expert field, so professional readers are more familiar to this term rather than CM54.

Thank you, we agree that Hu08 and CM54 are confusing terms. We also found that including the Cox-Munk parameterization with all of our results made the aim of the paper less clear, so we instead dedicate model comparison analysis to Figs. 7 and 8. When these parameterizations are mentioned, we simply say Hu and Cox-Munk instead of Hu08 and CM54.

- Equations are not common for the introduction of peer-reviewed studies, but I'll leave this decision up to the editors.

  Thank you for pointing this out. To lessen confusion, we removed this equation entirely.

- CALIPSO night data have higher quality from signal-to-noise ratio standpoint, but you said that only daytime data is available for AOD retrieval for correcting ocean signal (see Line 61). Clarify this aspect please.

  Thank you for bringing up this point. This aspect has been clarified as follows:

  Added: "If coincident aerosol optical depth (AOD) data are available (e.g., from MODIS in the case of CALIPSO detailed in Josset, 2008) then they may be used to estimate the intervening attenuation and transfer the calibration. However, such data from passive sensors including MODIS are only available during daytime, are typically not produced in the vicinity of clouds and may have unacceptably high uncertainties for accurately accounting for aerosol attenuation."

- The indicative paragraph of the introduction is poorly structured (the last paragraph of the introduction). A reader should get a clear idea about your research aim, but this paragraph might actually confuse a reader by excessive information inserted here. Please follow the structure: methodology shortly, research aim, site/period. All aspects requiring extensive referencing can be mentioned before this paragraph.

  Thank you, we agree that this paragraph is too lengthy. We revised this paragraph using the structure that you suggested (methodology shortly, research aim, site/period, sentence shortly summarizing main discussion points).

**Methodology:** This section should be revisited as well:

- Provide explicit and extensive description of your HSRL system before you introduce the concept of the ocean surface return retrieval. Without knowing the specifications of your HSRL system, it is impossible to judge whether the calculus you choose and, for instance, crude assumption about Fresnel reflectance (if we are speaking about non-nadir angles) is valid for your study or not.

  Thank you for this advice. We agree that it is important to introduce the HSRL-2 system before delving into the retrieval algorithm. Therefore, we created a new section: 2.3 HSRL-2 Instrument Description.

Added: "The NASA LaRC HSRL-2 is an airborne lidar instrument designed to enable vertically resolved retrievals of aerosol properties such as aerosol backscatter and depolarization at three wavelengths (355, 532, and 1064 nm), aerosol extinction at two wavelengths (355 and 532 nm) (Hair et al., 2008; Burton et al., 2018), and aerosol classification (Burton et al., 2012). In addition to these aerosol products, other retrieval capabilities include retrievals of atmospheric mixed layer height (Scarino et al., 2014), ocean subsurface particulate backscatter and attenuation coefficients (Schulien et al., 2017), cloud optical properties (in development), and 10 m surface wind speeds, the latter of which is the focus of this study. Details of the laser receiver optics and detectors are described in detail in Hair et al. (2008). This analysis utilizes the 532 nm data channels that include a total scattering channel (both molecular and particulate scattering), molecular scattering only, and the cross polarized channel, which are internally calibrated during flight. Key to determining the optical transmission and subsurface signals is a molecular channel that filters essentially all the particulate and specular scattering using the iodine notch filter as described in Hair et al. (2008), determining both the laser transmission down to the surface and correction of the subsurface scattering contribution to the integrated surface backscatter signal. The lidar is operated in a nadir-only viewing geometry (i.e., not scanning). The laser is a custom built 200 Hz repetition rate Nd:YAG laser emitting at 1064 nm, which is converted to both the second and third harmonic wavelengths of 532 nm and 355 nm, respectively. The output laser energies are nominally 34 mJ (1064 nm), 11 mJ (532 and 355 nm each) and each is set to a divergence ($1/e^2$) of approximately 0.8 mrad, giving a beam footprint diameter on the ocean surface of ~7 m for the nominal 9 km King Air flight altitude. The telescope is set to a full field of view of 1 mrad, giving a viewing footprint diameter of 9 m at the ocean surface at nominal flight altitude. All three wavelengths are transmitted coaxially with the telescope through a fused silica window in the bottom of the aircraft are actively boresighted to the receiver. The HSRL-2 incorporates high speed photomultiplier tubes (PMTs) and custom amplifiers to allow data collection at 120 MHz sampling rates with 40 MHz bandwidths. Data are sampled at 120 MHz (1.25 m in the atmosphere and 0.94 m in the ocean) with 16-bit digitizers and single-shot profiles are summed over 100 laser shots during 0.5 s which is the fundamental acquisition interval before storing to a disk. The aircraft incorporates an Applanix Inertial Navigation System (INS) to record the aircraft altitude at 0.5 s time intervals corresponding to each 100-shot data profile. For the surface wind speed calculations, data are screened to limit the pitch and roll to less than +/- 3° from the median values, which are approximately 0° for the roll and 3° - 5° for pitch on the King Air."

- From my point of view, the structure of your methodology should be: instrument, campaign, calculus, correction/collocation procedures. Now it's: "very short instrumental description, calculus, instrument a bit again, campaign, correction/collocation procedures, trivial statistical wiki", but it is up to you.

We appreciate your feedback about the structure of our Methods section. We reordered Section 2 as follows: campaign, dropsonde instrument description, HSRL-2 instrument description, HSRL-2 retrieval algorithm, collocation/statistical procedures.

- I think sub-section 2.5 is redundant for the journal like AMT. These statistical approaches are common knowledge for atmospheric research.

  Thank you for this comment. We decided to merge this sub-section with the collocation procedures (both are now Sect. 2.5), and removed the redundant definitions (i.e., equations for mean, STD, percentiles). We kept this section mainly because the other reviewer was quite interested in why we use OLS-bisector rather than other errors-in-variable techniques, so we expanded this discussion. We also wanted to be clear on which statistics we use (mainly for the mean error definition), which is why we provide a brief list at the end of the sub-section.

  Revised: "Since wind speeds are the focus of this study, first the dropsonde wind speed data points closest to 10 m (altitude of 11.56 m ± 3.19 m for the 577 points) above sea level are recorded for each launch (multiple launches per flight) to allow meaningful comparison with the HSRL-2 surface wind speeds. Since one data point was taken per dropsonde for each flight, there are 160 recorded dropsonde measurements for 2020, 245 measurements for 2021, and 335 measurements for 2022. Then, the HSRL-2 wind speed retrieval closest in space and time to the corresponding dropsonde measurement is recorded. Collocation between the HSRL-2 and the dropsondes is constrained to below 30 km horizontally and below 15 minutes temporally to remove outliers while trying to maximize the number of data points to be used in the study. Further constraining these distance and time conditions would eliminate more data points with negligible improvement to the statistics as shown by Figs. S1 and S2 in the supplement. Due to missing data in the HSRL-2 data set and the removal of outliers based on collocation constraints, 577 data points are available for comparison between the dropsondes and the HSRL-2 (Fig. 3).

[Figure]

**Figure 3: Map of 577 ACTIVATE dropsondes launched from the King Air between 2020 and 2022 that are used to evaluate the HSRL-2 surface wind speed retrievals introduced in this study.**

After the surface wind speed data are prepared using the procedure above, scatterplots along with the correlation coefficient (r), linear regression, and ordinary least squares bisector regression (OLS-bisector) are used to visually demonstrate how well HSRL-2 wind speed data match dropsonde data and show any potential variability in the data. Since OLS-bisector is less common than linear regression, a brief explanation of their differences is provided. In linear regression, X is treated as the independent variable while Y is treated as the dependent variable. In other words, one observes how Y varies with changes to fixed X values. OLS-bisector is known as an errors-in-variable regression technique, where X and Y are both dependent variables and thus both subject to error. OLS-bisector regresses Y on X (standard OLS) and then regresses X on Y (inverse OLS), then bisects the angle of these two regression lines (Ricker, 1973). Although other errors-in-variable techniques exist (e.g., Deming regression, orthogonal distance regression), OLS-bisector is chosen because it calculates the error present in both data sets using the bisector rather than assuming an error a priori like the examples mentioned (Wu and Yu, 2018). After performing these regressions, histograms of surface wind speed deltas, which are defined as HSRL-2 wind speed minus dropsonde wind speed, are created to show the distribution and spread of the data more easily. The mean and standard deviation (STD), of the surface wind speed deltas are computed and then used to define the mean error (mean ± STD). This metric is used to evaluate how accurately the HSRL-2 retrieves surface wind speeds. The mean and STD are then used to calculate the error (mean ± STD) of the HSRL-2 wind speed product. Note that mean and bias are used interchangeably in the following discussion."

- You mentioned MERRA-2 reanalysis (Lines 423-425), but never described this data in the methodology. If you really used this data, this is a critical oversight, making your study unreproducible and therefore not suitable for peer-review. If you did not use it, it's confusing why you suddenly show us SST background map here like it will be an important aspect of your study later.

  We apologize for the confusion. We simply use the MERRA-2 data to contextualize sea-surface temperature gradients and do not use any of its surface wind speed data for the comparison analysis. This data is just used to briefly show that we can potentially track changes in sea-surface temperature using HSRL-2 surface wind speed.

  Added: "Note that Fig. 3a uses SST data from Modern-Era Retrospective Analysis for Research and Applications, Version 2 (MERRA-2) (Gelaro et al., 2017) to contextualize the SST gradients present in the WNAO, and no comparisons with MERRA-2 surface wind speed data are performed in this study."

**Results:** I am afraid that the figures 1 – 8 and corresponding description alongside the tables are nothing but chaos from peer-review study point of view. Think about readers, would it be convenient for them to comprehend the material in this way? Please follow the structure "*Description of figure 1, Figure 1, then Description of Figure 2, Figure 2, … etc*" It does not mean you need to make every plot as a separate figure but speak out what you show us. Also,

Tables are wrongly formatted. See the AMT requirements for border formatting when it comes to tables. Also, please ensure that you report quantitative findings consistently without over-relying on qualitative terms.

Thank you, we agree that listing all the figures at once is quite confusing for the reader. We now have revised Section 3.1 with the following structure: introduce Figure X, Figure X, discussion of Figure X. We have also revised the table formatting to follow AMT requirements and report our quantitative findings from each figure.

Revised:

Table 1: Summary of all HSRL-2 – dropsonde surface wind speed statistics shown in Figs. 7 - 9. The two values for slope and y-intercept refer to those for the linear and bisector regressions, in that order. R values are the same for both linear and bisector regressions, so they are listed as one value.

| | N | r | Slope | Y-intercept [m s$^{-1}$] | Mean Error [m s$^{-1}$] |
|---|---|---|---|---|---|
| Overall | 577 | 0.89 | 1.04/1.17 | -0.13/-1.05 | 0.15 ± 1.80 |
| Wind Speed < 7 m s$^{-1}$ | 292 | 0.66 | 0.65/0.99 | 1.10/-0.49 | -0.54 ± 1.34 |
| 7 m s$^{-1}$ ≤ Wind Speed < 13.3 m s$^{-1}$ | 236 | 0.75 | 0.64/0.85 | 3.80/1.87 | 0.56 ± 1.49 |
| Winter | 236 | 0.88 | 0.95/1.08 | 1.03/-0.08 | 0.63 ± 2.07 |
| Summer | 341 | 0.87 | 1.08/1.24 | -0.69/-1.68 | -0.18 ± 1.52 |

**Conclusions:** This is a section with little value for reader in the current form. Please revisit your conclusions by thinking about two aspects: harmonizing this section of the manuscript with key aspects such as research aim, methodology, key results, etc; second, think about implications you give for future studies. Good and bad lessons are both valuable, use them in the best way to inform the reader. Specifically:

- Foremost, I think you should report your correlations for each wind speed range directly. I see correlations only over the entire ranges, which might be misleading due to highly variable correlations depending on the wind speeds. Stick to quantitative reporting please without switching to qualitative remarks, behind which, a reader cannot discern an actual low correlation for high wind speeds. Such correlations are also key, essential lessons learned from your analysis, making your study more valuable.

  Thank you for this great suggestion. We agree that there is a lot of value in discussing each wind speed range directly and quantitatively in the conclusions, so we include this aspect now.

  Added: "It is also observed that the dropsonde surface wind speed measurements most closely match with the Hu et al. (2008) wind speed – wave-slope variance model than the Cox and Munk (1954) and Wu (1990) models for surface wind speeds below 7 m s$^{-1}$, which is an important finding because most ACTIVATE surface wind speeds fall into this category. After this overview of model performance, the HSRL-2 retrievals for surface wind speeds separated into below 7 m s$^{-1}$ and between 7 m s$^{-1}$ and 13.3 m s$^{-1}$ categories are

then evaluated in more detail. For surface wind speeds below 7 m s$^{-1}$, correlations of 0.66, slopes of 0.65 and 0.99, and y-intercepts of 1.10 m s$^{-1}$ and -0.49 m s$^{-1}$ are found and the accuracy of the retrievals is found to be -0.54 m s$^{-1}$ ± 1.34 m s$^{-1}$. Surface wind speeds between 7 m s$^{-1}$ and 13.3 m s$^{-1}$ show correlations of 0.75, slopes of 0.64 and 0.85, and y-intercepts of 3.80 m s$^{-1}$ and 1.87 m s$^{-1}$ and the retrieval accuracy is shown to be 0.56 m s$^{-1}$ ± 1.49 m s$^{-1}$. Statistics are not reported for surface wind speeds above 13.3 m s$^{-1}$ because there are too few points in this category to make meaningful comparisons. These results showcase an important observation not seen in the cumulative results, which is that the HSRL-2 estimates surface wind speeds with a bias of ± ~0.5 m s$^{-1}$ and an error of ± ~1.5 m s$^{-1}$. Lastly, the data are divided into winter and summer deployments (dates denoted in Sect. 2.1) to assess how the HSRL-2 performs between seasons. The winter surface wind speed data comparisons show correlations of 0.88, slopes of 0.95 and 1.08, and y-intercepts of 1.03 m s$^{-1}$ and -0.08 m s$^{-1}$ and the summer data show correlations of 0.87, slopes of 1.08 and 1.24, and y-intercepts of -0.69 m s$^{-1}$ and -1.68 m s$^{-1}$ (linear and bisector regressions, respectively). The accuracy of the lidar retrievals is reported as 0.63 m s$^{-1}$ ± 2.07 m s$^{-1}$ and -0.18 m s$^{-1}$ ± 1.52 m s$^{-1}$ for winter and summer, respectively. These findings show that HSRL-2 retrievals are more accurate in the summer than in the winter, but still provide substantial (N = 236) and accurate surface wind speed data in winter as well."

- I suggest to remove a remark on the comparison between the efficiency of CM-54 and Hu-08 models because you never set up such research aim; there was no ultimate analysis on this, right? However, note that some other expert might thinks, well, once the authors state that they can actually ESTIMATE surface reflectance, why they did not go all the way by comparing Cox-Munk, Wu 90, Hu 08, Li-2010 etc, parametrizations QUANTITATIVELY if they finally get their hands on the ground truth using HSRL when it comes to sea surface reflectance?

Thank you, we greatly appreciate your advice on this. We instead perform a preliminary model analysis in Figs. 7 and 8 to show that the Hu model performs better than the Cox-Munk and Wu models for surface wind speeds below 7 m s$^{-1}$. A more rigorous analysis as you mentioned is warranted, but the aim of our paper mainly is to evaluate our retrieval algorithm using the in situ dropsonde measurements. This detailed efficiency analysis would be good material for a second manuscript, but is beyond the scope of this paper.

Added: "Now that the HSRL-2 retrievals have been broadly evaluated, Fig. 7 shows how their accuracy varies per 1 m s$^{-1}$ interval in surface wind speed. This plot also provides the opportunity to compare the Hu et al. (2008) model with the models proposed by Cox and Munk (1954) and Wu (1990) to see if some of the error in the HSRL-2 retrievals can be attributed to model characteristics.

[Figure]

**Figure 7: HSRL-2 surface wind speed using Hu, Cox-Munk, and Wu models versus mean dropsonde surface wind speed calculated per 1 m s$^{-1}$ bin. A histogram of dropsonde surface wind speeds is also included to show their distribution.**

It is seen that the mean Cox-Munk and Wu surface wind speed values are higher than the mean Hu values from 0 m s$^{-1}$ to 7 m s$^{-1}$, showing that the Cox-Munk and Wu relationships overestimate dropsonde surface wind speeds more than the Hu relationship. The variability (i.e., STD) around the mean per bin is similar between the three models, which is 1.59 m s$^{-1}$ for Hu, 1.43 m s$^{-1}$ for Cox-Munk, and 1.55 m s$^{-1}$ for Wu on average. Although similar, the STD of the Hu surface wind speeds found here is ~0.4 m s$^{-1}$ lower than the one found in Fig. 6. This could be attributed to an STD not being able to be calculated for the 17 to 18 m s$^{-1}$ bin since it only contained one point.

Although it is apparent Cox-Munk and Wu retrievals overestimate dropsonde observations for surface wind speeds below 7 m s$^{-1}$, it is still unclear which of the models perform better overall. Therefore, the y-axis from Fig. 7 is converted to wave-slope space and the result of this modification is shown in Fig. 8. HSRL-2 wave-slope is used because it directly reports the original measurements of surface reflectance rather than estimated values of surface wind speed. Using the original data ensures that uncertainty is coming from the actual HSRL-2 – dropsonde comparisons rather than from potential errors in the conversion from wave-slope to surface wind speed.

[Figure]

**Figure 8: HSRL-2 wave-slope variance versus mean dropsonde surface wind speed calculated per 1 m s⁻¹ bin. Ideal Hu, Cox-Munk, and Wu distributions are included to show how well observed dropsonde data match with each parameterization. A histogram of dropsonde surface wind speeds is also included to show their distribution.**

From Fig. 8, it is more easily seen how the dropsonde surface wind speed distribution compares with Hu, Cox-Munk, and Wu parameterizations. Dropsonde surface wind speeds match quite closely to Hu and Cox-Munk parameterizations as opposed to the Wu parameterization between 7 m s⁻¹ and 13.3 m s⁻¹, although some divergence is seen above ~10.5 m s⁻¹. However, a critical observation that is more apparent in Fig. 8 than Fig. 7 is how the dropsonde data most resemble the Hu distribution for surface wind speeds below 7 m s⁻¹. This improvement is substantial, especially since most of the surface wind speeds in ACTIVATE fall into this category. Surface wind speeds above 13.3 m s⁻¹ substantially diverge from all models, especially above 16 m s⁻¹. As mentioned previously, there are few surface wind speed observations in this category, so more measurements are necessary to make meaningful comparisons between the two data sets. Overall, Figs. 7 and 8 demonstrate the benefits of using the Hu parameterization in this study and why surface wind speeds above 13.3 m s⁻¹ are not the main focus of the comparisons in this section. Further analysis is warranted to rigorously compare the performance of various surface reflectance models and potentially apply corrections (i.e., whitecap correction for surface wind speeds above 13.3 m s⁻¹), but the aim of this paper is to evaluate LARC's HSRL-2 surface wind speed retrieval algorithm using the available ground-truth dropsonde measurements."

Line 496. A very strange remark appeared here. You used the word "novel" for the first time in the conclusions, you never ever articulate it before the conclusive remark is made. Which leads to conclusion, have you been surprised that your method is novel once you reached conclusions? Explain the novelty of your method in a comment below and provide required explanations in abstract as well.

Thank you for pointing this out. What we meant is that this is a new ocean surface wind speed product that LaRC's HSRL-2 team developed, so this is now reflected throughout the paper. As a result, we remove the "word" novel from this line.

Revised: "This retrieval method offers a new path forward in airborne field work for the acquisition of surface wind speed data at a high spatial (~75 m along track) and time (0.5 s) resolution, as demonstrated with two case study flights (Research Flight 29 on 28 August 2020 and Research Flight 14 on 1 March 2020)."

Minor comments:

- Line 64 Errors can create errors is a bad word choice.

  We agree. Line now reads: "Estimation of the attenuation from the lidar data alone requires an assumption of the aerosol extinction-to-backscatter ratio (or "lidar ratio"), so errors in the assumed value can lead to an incorrect estimate of attenuation, especially when AOD is high."

- Line 99 Section 1 = Introduction

  This line has been removed entirely to decrease the length of the indicative paragraph of the Introduction.

- Line 104 This approach, as well as Hu-08 piece-wise approach for U derivation; both will not work with non-nadir lidar systems (see works of Josset et al 2010; about the equation for non-nadir retrieval of ocean surface reflectance and ongoing works of Labzovskii et al., on non-nadir retrievals of ocean surface reflectance from Aeolus which measures at >35 degree incidence).

  Thank you for pointing this out. We add that we use a nadir-only viewing geometry in the instrument description (Sect. 2.3), which is why we use the approaches of Josset et al. (2010b) and Hu et al. (2008).

  Added: "The lidar is operated in a nadir-only viewing geometry (i.e., not scanning)."

- Line 108 Is your system similar to CALIPSO to assume this Hu approach works with your HSRL system?

  Thank you for this question. We assume that we can use the Hu approach with the HSRL-2 system because the measurement channels we introduce in Eqs. 4.1 and 4.2 are said to

be similar to CALIPSO's 532 nm channels in that they both measure the attenuated backscatter from molecules and particles. However, there is a limitation in using the Hu relationship for surface wind speeds above 13.3 m s$^{-1}$ due to some significant differences between CALIPSO and HSRL-2, which is discussed in the whitecaps section at the end of Sect. 2.4.

Added: "The equations for the HSRL-2 532-nm measurement channels are:

[revised manuscript text omitted]

- Line 115 Structural setback of your methodology is evident here. You talk about HSRL, then calculus, then HSRL again. Can you first extensively discuss your HSRL system and only then, to justify that Hu/Josset/… etc approaches demonstrated for CALIPSO are applicable in your case?

  Thank you, we extensively discuss the HSRL-2 instrumentation in Sect. 2.3 (paragraph added in previous comment about providing extensive description of the HSRL-2 instrument) as you suggested to show that we can apply the CALIPSO methodology to the HSRL-2.

- Line 128. Broken out = separated?

  Thank you, we changed "broken out" to "separated" for clarity.

- Line 159 "Arb units"?

  Thank you for pointing this out. We've revised Fig. 1 to say $m^{-1}\ sr^{-1}$ instead of "arb units".

  Revised:

[Figure]

**Figure 1: Visualization of HSRL-2 measurement signals as described in Eqs. 6 – 8. Dashed line denotes ideal total backscatter signal from the atmosphere, surface reflection, and the ocean subsurface. Blue and black lines denote measured signals from total and molecular scattering channels, respectively. Red and green lines show the ocean corrected signal and the ocean surface backscatter, respectively. Dots indicate the altitudes of digitized samples. The sampling rate is 120 MHz, resulting in a vertical spacing of 1.25 m in the atmosphere and 0.94 m in the ocean.**

- Line 165 Please address this structural setback here and elsewhere in the manuscript. In peer-reviewed studies you normally first describe the figure, and then show it below.

We appreciate this advice. We moved the sentence "Figure 1 illustrates the vertical distributions of the measured signals $P_{tot}$ (black) and $P_{mol}$ (blue) along with the $P_{tot}^{surf}$ (green) component of $P_{tot}$. Note that zero altitude is the location of the ocean surface." before Fig. 1 and followed this structure throughout the rest of the figures.

Revised: "Figure 1 illustrates the vertical distributions of the measured signals $P_{tot}$ (black) and $P_{mol}$ (blue) along with the $P_{tot}^{surf}$ (green) component of $P_{tot}$. Note that zero altitude is the location of the ocean surface.

[Figure]

**Figure 2: Visualization of HSRL-2 measurement signals as described in Eqs. 6 – 8. Dashed line denotes ideal total backscatter signal from the atmosphere, surface reflection, and the ocean subsurface. Blue and black lines denote measured signals from total and molecular scattering channels, respectively. Red and green lines show the ocean**

corrected signal and the ocean surface backscatter, respectively. Dots indicate the altitudes of digitized samples. The sampling rate is 120 MHz, resulting in a vertical spacing of 1.25 m in the atmosphere and 0.94 m in the ocean.

It is seen from Fig. 1 and Eq. 8b that the surface component $P_{tot}^{surf}$ of the measured signal $P_{tot}$ is not localized to the surface but is instead spread above and below the surface via convolution with the system response function..."

- Line 170 Here, your structural setbacks become critical, please split your methodology into: description of HSRL and explanation of the calculus + if needed, correction procedures for HSRL.

  Thanks for the suggestion to include an instrument description and we agree. We have added a short instrument description that gives the relevant information of the instrument and its configuration in Sect. 2.3 as mentioned in previous comments and then transition into the retrieval methodology.

- Line 175. Not true, it also depends on the wavelength and the incidence angle of lidar (see works of Li et al., 2010 on pre-launch Aeolus demonstrator; Josset et al., 2010 work on non-nadir retrieval of Beta_surf and Labzovskii et al. ongoing Aeolus works on applying this principle to Aeolus setup with UV wavelength and >35 angle incidence). Add references if needed for justification and readers' interest.

  The magnitude of the subsurface scattering (i.e., hydrosol scattering) within the integration window around the surface does rely on the amount and type of the particulates in the ocean. The magnitude of the subsurface scattering does depend on the wavelength and to a lesser extent the angle. We also agree that the surface scattering will depend on the angle of incidence and the wavelength. As suggested by both reviewers, we added a short description of the HSRL-2 instrument (paragraph shown in previous comments) that provides the specific geometry implemented on the aircraft and the wavelength used for this analysis. As now noted, the system has three wavelengths (355, 532, and 1064 nm) and we are using 532 nm due to the unique capabilities of the receiver filter to separate the Mie scattering signal accurately. The system is not scanning and points nominally nadir as clarified in a previous comment. The aircraft altitude dictates the angular changes relative to the surface. To limit rapid changes in the aircraft altitude, we have limited the angles of the pitch and roll to be less than 3° from the nominal values during straight and level legs. This will capture the level legs of the flight and limit turns where the angles change rapidly over the 0.5 s data integration times.

- Line 176. Once again, this assumption depends on the setup of your lidar system which is a mystery after the incomplete methodological description. Not true for non-nadir systems (sensitivity starts from U = 12 m/s if your incidence is > 30 degrees according to Aeolus pre-launch works of Li et al. 2010 for instance).

  Thank you, we hope that our added instrument description (shown in previous comment) clears this line up.

- Line 180. This is unsupported surmise because (a) we do not see any results, confirming this statement, (b) nor a reference, where a reader can get familiarized with this common expert knowledge.

We have reworded this sentence to be clearer based on your comment. The statement simply refers to the ocean subsurface contribution that would contribute to the total integrated backscatter signal if not corrected. This is simply stating that a larger attenuation in the ocean results in less contribution of the integrated signal. The intent of including this is to highlight that the subsurface contribution is higher for clear water compared to more turbid regions and results in a larger bias if not accounted for in the calculation of the surface backscatter. In summary, this simply states that the faster the ocean subsurface signal decays, the contribution becomes less which is the blue shaded region in Fig. 1.

Revised: "Therefore, the ocean subsurface contribution is higher for clear water compared to turbid water. For example, in the case illustrated in Fig. 1, the seawater particulate and molecular scattering are equal, resulting in a contribution of only 3.8% to the integrated surface backscatter as compared to the no particulate scattering noted above of 5.7%. The atmospheric signal contribution is much less (~100 times smaller) than the ocean subsurface signal and therefore its contribution is considered negligible. Fortunately, the high vertical resolution of the HSRL-2 instrument enables the ocean subsurface contribution to be estimated."

Line 188 Do you need to account for aerosol transmittance as well or you mean only molecular transmittance?

Thanks for the comment on being clear here. This accounts for the total attenuation but is determined from the molecular channel.

Revised: "The two-way total (particulate and molecular attenuation) transmission..."

- Line 198 "Unique to the HSRL-2…" rephrase this sentence please, it is ambiguous in the current form. Below, "to do so" -> replace to any structure, more common for academic English.

Thank you for this advice on grammar.

Revised: "A benefit of the HSRL-2 retrieval algorithm is that one can use the molecular channel signal to determine the ocean signal near the surface (see Fig. 1). To determine the near-surface ocean signal, an estimate of the total ocean scattering ratio (TSR) is employed, which is the ratio of molecular + hydrosol backscatter divided by molecular backscatter."

- Line 205 Is it actually true, can TSR be constant in real conditions over considerable spatio-temporal range??

Thanks for the comment and we have made some clarifying comments on this assumption. From the lidar measurements, one can only assess the homogeneity near the surface (> 5 m) due to the fact that the surface contribution is present within 5 m of the surface. We are

only requiring it to be vertically homogeneous from the surface to the depth (5 - 8 m), where the TSR is calculated. This is difficult to assess even with in situ backscatter measurements done aboard ships. However, given the ocean mixed layers are generally > 8 m in the open oceans, this assumption is reasonable for the small (0 - 6% contribution to the integrated signal).

Revised: "Here the assumption is that the TSR is vertically constant near the surface over the 0.5 s (~75 m horizontal resolution) integration of the lidar signals.".

- Line 213 Ideal -> replace to some other word or justify why it is "ideal" here.

    Revised: "The HSRL-2 ocean surface wind speed product is assessed during the ACTIVATE campaign, which is a NASA Earth Venture Suborbital-3 (EVS-3) mission."

- Line 380 I think you mean that CM54 and Hu08 demonstrate high bias, no? Also, why to report all-range-encompassing results, while you literally made piece-wise detailed analysis for every range? Please explain. I think this without this aspect, you can more clearly report bias/correlations/any other issues between HSRL and dropsondes right in your abstract as well without being confused to make additional comments on higher or lower correlation at some wind speed range. I hope my comment is clear here.

    Thank you for this comment. We now clearly report the statistics of the 1) cumulative, 2) wind speed range, and 3) seasonal results in the abstract as mentioned in a previous comment. The wind speed range results show that the HSRL-2 retrievals underestimate surface wind speeds for those below 7 m s$^{-1}$ and overestimate them for those in between 7 m s$^{-1}$ and 13.3 m s$^{-1}$, which is an important result that was not reflected in the previous manuscript.

- Line 395 What is winter and summer for you?

    We apologize for this confusion. We define these definitions at the end of the first paragraph in Sect. 2.1: "Winter deployments included the following date ranges: 14 February – 12 March (2020), 27 January – 2 April (2021), 30 November 2021 – 29 March (2022). Summer deployments were as follows: 13 August – 30 September (2020), 13 May – 30 June (2021), 3 May – 18 June (2022)."

    The seasonal deployments tended to take place in different months year to year, which is why we list it this way.

- Lines 400… I did not see where you discuss the complete lack of correlation at high wind speeds seen at Figure 8b for instance. Can you navigate me please?

    Thank you for this comment. We now remark starting in the discussion of Fig. 8 that dropsonde surface wind speeds above 13.3 m s$^{-1}$ substantially differ from Hu, Cox-Munk, and Wu wind speed - wave-slope distributions. Also, there are few points in this category, so it is difficult to make meaningful comparisons between the HSRL-2 retrievals and the

dropsonde measurements. Therefore, we no longer show comparisons in the Results section due to the complete lack of correlation for such few points.

Added: "Surface wind speeds above 13.3 m s$^{-1}$ substantially diverge from all models, especially above 16 m s$^{-1}$. As mentioned previously, there are few surface wind speed observations in this category, so more measurements are necessary to make meaningful comparisons between the two data sets. Overall, Figs. 7 and 8 demonstrate the benefits of using the Hu parameterization in this study and why surface wind speeds above 13.3 m s$^{-1}$ are not the main focus of the comparisons in this section."

- Line 424. Which day, can you reiterate in the text as well for convenience?

  Thank you, 1 March 2020 is now reiterated for clarity and convenience: "This flight along with the associated morning flight on 1 March 2020 have been the subject of several studies owing to its coincidence with cold air outbreak conditions."

- Line 427 (and also 440, 456). "Significant" is a statistical term for peer-reviewed studies; it requires some arguments on statistical significance. If you did not mean that, use "substantial" as your word instead. Also, add a reference about common surface temperature gradient in this area. Think about general readers and argumentation, please.

  Thank you, we changed instances of "significant" to "substantial" throughout the paper. Also, Painemal et al. (2021) is added to show what common SST gradients in the WNAO look like.

  Added: "These conditions allow for the examination of how the high horizontal spatial resolution of the HSRL-2 (~75 m along track as mentioned in Sect. 2.4) influences its retrievals and how the data can be used to track sea surface temperature (SST) gradients common to the WNAO (Painemal et al., 2021) as seen in Fig. 3."

- Line 430 Sounds like this flight should have been described in the methodology or?

  We decided to explain this flight here because we thought these details were too specific for the ACTIVATE mission description. We also decided against a Case Study section in the Methods since these flights are more for context on potential uses of the surface wind speed data. Therefore, no changes to the Methods are made concerning the case studies.

- Line 436 Explain to a general reader which physical principle is the fundament of this HSRL ability please. Mention actual spatial resolution which is used to resolve such gradient.

  This is an important point. We now add that the 0.5 s or ~75 m along-track resolution is what allows us to resolve these gradients first in the 28 August 2020 research flight discussion and throughout the Case Study section in general.

Added: "These conditions allow for the examination of how the high horizontal spatial resolution of the HSRL-2 (~75 m along track as mentioned in Sect. 2.4) influences its retrievals and how the data can be used to track air-sea interaction dynamics such as sea surface temperature (SST) gradients as seen in Fig. 3... these observations show that the HSRL-2 has the high horizontal spatial resolution needed to probe the fine-scale variability of surface wind speeds and has the potential to improve atmospheric modeling of MABL processes."

- Line 443 One would argue that 3.32 m s-1 is not agreement but discrepancy, e.g. substantial bias, no?

  Thank you, this comment along with others inspired us to restructure the 28 August 2020 case study to show what 1 March 2020 originally was intended to communicate. Instead of focusing on the biases seen between the collocated points, we wanted to show how we can use the HSRL-2 surface wind speed data in general and how the high spatial and time resolutions allow us to track changes in surface winds over time throughout a flight day.

- Line 447. Yes, it might be difficult during high cloud fraction conditions. Thus, this method is also constrained like Hu et al. 2008 method relying on clear atmospheric conditions for retrieving surface reflectance from ocean?

  Based on this comment and feedback from Reviewer 2, we removed the 11 January 2022 case study and use 1 March 2020 to show that the HSRL-2 can still make some retrievals on days with broken cloud scenes. Therefore, we show that we do not need to rely on aerosol- and cloud-free conditions exclusively like in Hu et al. (2008) for the HSRL-2 to retrieve surface reflectance.

  Added: "Next, Research Flight 14 is shown in Fig. 5 to demonstrate the ability of the HSRL-2 to sample in broken cloud scenes. This flight along with the associated morning flight on 1 March 2020 have been the subject of several studies owing to its coincidence with cold air outbreak conditions (see cloud streets in Fig. 5a) and a flight strategy that allowed for detailed characterization of the evolving aerosol-cloud system as a function of distance offshore (Seethala et al., 2021; Chen et al., 2022; Li et al., 2022; Tornow et al., 2022; Sorooshian et al., 2023). The morning flight focused on a location with very detailed characterization including stacked level flight legs (i.e., termed a "wall") with the Falcon flying below, in, and above clouds, with the King Air flying aloft to further characterize the same region. The afternoon flight consisted of both aircraft flying back to that same location, adjusting the sampling strategy to fly along the boundary layer wind direction in a quasi-Lagrangian fashion to keep studying the evolution of the air mass characterized in the morning. The afternoon flight is chosen because it shows the full range of cloud conditions from clear to completely overcast. Therefore, the HSRL-2 surface wind speed retrievals are able to be evaluated in this range of conditions.

[Figure]

**Figure 5: a) Flight map of the King Air (red line), Falcon (yellow line), and dropsondes (dark yellow circles) overlaid onto Geostationary Operational Environmental Satellite (GOES-16) cloud imagery for Research Flight 14 on 1 March 2020. Blue stars represent time stamps where the King Air crosses over from cloud-free to cloudy areas. b) Time series of surface wind speed data from HSRL-2 and dropsondes for the same flight, where lines signify total HSRL-2 surface wind speed data and circles indicate collocated surface wind speed data points. Blue dashed lines represent time stamps of interest as indicated in a).**

As the aircraft approaches the cloud scene at 19:18, there is a noticeable and steady increase of HSRL-2 surface wind speeds. The reverse observation is seen when the aircraft approaches 21:15, where the HSRL-2 surface wind speeds start to decrease steadily. As highlighted in the 28 August 2020 case study, the high horizontal spatial resolution of the HSRL-2 retrievals enables these spatial gradients to be observed. Another important takeaway is the HSRL-2 is still able to sample the surface in cloud scenes, as seen by the almost complete surface wind speed profile in Fig. 5b. Although a gap in data occurs at 20:15 where cloud cover is most substantial, some retrievals are still present in that area. The reason is that the HSRL-2 can probe the surface through gaps between clouds, allowing for the surface wind speed retrievals to take place. Although the HSRL-2 retrievals would be unavailable in overcast cloud scenes, the ability of the instrument to sample the surface in broken cloud fields and not just aerosol- and cloud-free scenes is a significant benefit of the lidar and the HSRL technique."

- Line 477 Perhaps, you might be even more assertive here? Do you think we really can use HSRL-2 wind speed retrievals in such conditions? I mean fair, transparent recommendation would work the best for your own benefit here from my point of view. This is important lesson learned, very valuable.

  We agree that we needed to be more explicit on whether the HSRL-2 could make retrievals on cloudy days, which is why we refocused the 1 March 2020 and removed 11 January 2022 like we mentioned in the previous comment (discussion added in previous comment). We now say that the HSRL-2 method can detect the surface in between clouds, showing that it does not rely exclusively on cloud-free conditions for the retrieval to work.

- Line 485/6 "Results being compared", can you simplify the wording here and elsewhere in conclusions please. Scientific analysis encourages brevity and clarity over vagueness and wordiness.

Thank you, we agree that the wording in the conclusions should have been more simplified to promote clarity. The "results being compared" sentence has been removed, but the entire Conclusions section is revised to be more succinct.

[revised manuscript text omitted]

---

## Author Comment (AC2)

We thank the reviewer for their thoughtful suggestions and constructive criticism that have helped us improve our manuscript. Below, we provide responses to reviewer concerns and suggestions in blue font.

Review of egusphere-2023-1943: "HSRL-2 Retrievals of Ocean Surface Wind Speeds" by Dmitrovic et al.

In this work the authors describe an algorithm for deriving ocean surface wind speed estimates from measurements of ocean surface backscatter acquired by the NASA-Langley high spectral resolution lidar (HSRL). The NASA HSRL can accurately characterize the signal attenuation above the ocean surface and reliably partition the surface signal pulse into pure surface and ocean subsurface components, and hence can deliver high quality measurements of surface integrated attenuated backscatter ($\beta_{surf}$). The accuracy of the wind speed retrieval thus depends on the equation relating $\beta_{surf}$ to wave slope variances and the fidelity of the model used to convert wave slope variances to wind speeds. The authors derive wind speeds using two different models – the classic Cox-Munk (1954) and a lidar-specific model developed by Hu et al. (2008) – and compare these results to near-simultaneous dropsonde measurements of wind speeds. The paper is well organized and well written and its subject matter is entirely appropriate for Atmospheric Measurements Techniques. There are, however, several issues that should be addressed prior to publication.

My primary concern is that the authors' equation (2) fails to acknowledge the possible presence of whitecaps and/or sea foam. While the authors cite Josset et al., 2010b as their source for equation (2), that work explicitly includes reflection from the whitecap fraction within any footprint; see section 2.2 and equations (2) and (21) therein. Furthermore, comparing the upper and lower panels of figure (2) in Hu et al., 2008 suggests that omitting whitecap contributions could have a significant impact on the results derived in this paper. On the other hand, Lancaster et al. (2005) suggest that the "contribution of whitecaps to the nadir lidar measurements is seen in Figure 2 to be negligible".

You bring up an excellent point and as noted in Josset et al. (2010b) there needs to be more research on the effects of whitecaps. Josset notes, "There is also a need to better assess the large uncertainties associated with the lidar return of foam patches and their effect on subsurface lidar returns." Also, Hu et al. (2008) provided new insight to this contribution by using the integrated surface depolarization ratio to add an empirical correction to the surface scattering of whitecaps. The details of how this correction was determined were not provided and there are significant differences between our lidar and CALIPSO. These include the spot size of the beam on the surface (8 m compared to ~90 m for CALIPSO and HSRL-2 has data collected and stored over 100 laser shots (averaged due to x10 repetition rate of the laser), while CALIPSO records every shot. We note that the CALIPSO data presented by Hu et al. (2008) was averaged globally and thus were statistically dominated by clear water cases where the change in depolarization would better

correlate with the whitecaps rather than the subsurface particulate scattering. Moreover, the data collected from ACTIVATE includes significant sampling along the coastal waters where the subsurface contribution from ocean particulates can significantly impact surface depolarization data. Lastly, the residual depolarization due to the different set of optics will likely change the empirical relationship compared to CALIPSO.

While recognizing that this correction is important at high wind speeds ($> 10$ m s$^{-1}$), we limited our discussion to the other two main factors in deriving wind speed using HSRL-2/1) the subsurface contribution due to the high vertical resolution to derive the ocean backscatter and 2) determining an accurate value of the atmospheric attenuation. It is of high interest to look at the surface depolarization, which we have calculated from HSRL-2 during these flights. There is a clear relationship between the surface depolarization with wind speed as expected, but the ocean subsurface contribution is also evident and correlates with increased scattering in the ocean. In addition, the minimum surface backscattered reflection could provide additional information on the average reflection of the whitecaps. For instance, looking at the 97th percentile the surface backscatter is 0.02. A method to account for the ocean subsurface contribution to the integrated depolarization would need to be accounted for in the retrievals. Critical to addressing all of these issues, data collected at higher wind speeds with correlative data is required. We note that Hu et al. (2008) had AMSR-E data with a large number of matchups in clear air and likely oligotrophic (low particulate scattering) ocean conditions. Therefore, we believe that the whitecap correction for HSRL-2 is not ready for publication without further analysis and evaluation, which has started but is still in the early phase.

We have added a general discussion about whitecap correction at the end of Sect. 2.4.

Added: "In addition to the specular reflection from the surface, whitecaps or sea foam can increase the lidar backscatter signal. As noted in Josset et al. (2010b), the contribution of scattering by the whitecaps on the ocean surface has been treated as Lambertian scattering. There is a wavelength dependence of the scattering at longer wavelengths due to the water absorption, based on measurements presented by Dierssen (2019) covering wavelengths from 0.4 – 2.5 µm. Measurements presented here are at 532 nm, a region of the visible spectrum where scattering from foam is relatively constant with wavelength. The contribution of whitecaps is typically modeled with a constant average reflectance and an effective area weighted fraction that varies with surface wind speed (Whitlock et al., 1982; Koepke, 1984; Gordon and Wang, 1994; Moore et al., 2000). Following Moore et al. (2000), we have estimated the average reflectance due to the whitecaps as a function of surface wind speed and the difference becomes $> 1$ m s$^{-1}$ for surface wind speeds $> 15$ m s$^{-1}$ based on this relationship. As presented below, there are limited data (49 data points) above 13.3 m s$^{-1}$ that can be compared to the dropsonde surface wind speeds to evaluate this relationship. Moreover, since the correction depends on surface wind speed, an iterative calculation is required to use this relationship as the backscatter is dependent on wind speed.

[Figure]

**Figure 2. Estimated absolute difference in calculated surface wind speed if reflectance from whitecaps is not included. The lidar surface backscatter is higher than the specular reflectance if whitecaps are present, which results in a lower estimated surface wind speed if not accounted for in the retrieval.**

Alternatively, Hu et al. (2008) used a full month of CALIPSO integrated surface depolarization ratio (ratio of the integrated cross polarized channel to the integrated co-polarized channel across the surface) and applied an empirical correction to the reflectance that was determined using AMSR-E data as the ground-truth data set to increase the correlation of the data sets. The correlation was based on much more data than the ACTIVATE matchups between HSRL-2 and dropsondes, limiting the utility of a similar analysis with the HSRL-2. In addition, there are significant differences in the configurations of CALIPSO and HSRL-2 that limit implementation of the same empirical relationship. First, CALIPSO's integrated surface depolarization includes the subsurface contributions due to its 30 m vertical resolution, whereas the HSRL-2 surface depolarization is integrated over only a few meters as shown in Fig. 1. Second, the CALIPSO data is based on global data, which is dominated by oligotrophic (clear) waters, whereas a significant fraction of the HSRL-2 - dropsonde comparisons are from eutrophic and mesotrophic waters near the coast and along the shelf. Third, there is a significant difference in footprint size between HSRL-2 and CALIPSO (8 m versus 90 m), with HSRL-2's instantaneous footprint area being greater than 2 orders of magnitude smaller and, considering HSRL-2's along-track averaging (100 laser shots) compared to CALIPSO's single shot data, greater than one order of magnitude smaller in terms of area over which surface depolarization is integrated."

References

Gordon, H. R. and Wang, M.: Influence of oceanic whitecaps on atmospheric correction of ocean-color sensors, Appl. Opt., 33, 7754-7763, 10.1364/AO.33.007754, 1994.

Dierssen, H. M.: Hyperspectral Measurements, Parameterizations, and Atmospheric Correction of Whitecaps and Foam From Visible to Shortwave Infrared for Ocean Color Remote Sensing, Frontiers in Earth Science, 7, 10.3389/feart.2019.00014, 2019.

Josset, D., Zhai, P.-W., Hu, Y., Pelon, J., and Lucker, P. L.: Lidar equation for ocean surface and subsurface, Opt. Express, 18, 20862-20875, 10.1364/OE.18.020862, 2010b.

Koepke, P.: Effective reflectance of oceanic whitecaps, Appl. Opt., 23, 1816-1824, 10.1364/AO.23.001816, 1984.

Moore, K. D., Voss, K. J., and Gordon, H. R.: Spectral reflectance of whitecaps: Their contribution to water-leaving radiance, Journal of Geophysical Research: Oceans, 105, 6493-6499, https://doi.org/10.1029/1999JC900334, 2000.

Whitlock, C. H., Bartlett, D. S., and Gurganus, E. A.: Sea foam reflectance and influence on optimum wavelength for remote sensing of ocean aerosols, Geophysical Research Letters, 9, 719-722, 1982.

Hu, Y., Stamnes, K., Vaughan, M., Pelon, J., Weimer, C., Wu, D., Cisewski, M., Sun, W., Yang, P., Lin, B., Omar, A., Flittner, D., Hostetler, C., Trepte, C., Winker, D., Gibson, G., and Santa-Maria, M.: Sea surface wind speed estimation from space-based lidar measurements, Atmos. Chem. Phys., 8, 3593-3601, 10.5194/acp-8-3593-2008, 2008.

I note that the authors' equation (2) also omits the atmospheric two-way transmittance term give in equation (21) in Josset et al., 2010b. Given the discussion in the introduction about calibration transfer and the assertion on line 116, this omission seems a bit surprising.

Thanks for the comment since it highlights the need to be clearer on this point. Although correct in Josset et al. (2010b), the formulation is given as the attenuated backscattered signal, which includes the atmospheric attenuation as pointed out in your comment. Here we provide the backscattered (180°) reflected radiance (units $sr^{-1}$) of the incident light level just above the surface as this is the quantity of interest and is directly related to the wave-slope variance and therefore wind speed. We have changed this from surface backscatter to surface backscatter (180°) reflected radiance. Note that the attenuation in our formulation is introduced when the signals from the lidar are included as shown in Eq. 9.

Added: "To derive surface wind speeds, the surface backscattered (180°) reflected radiance ($\beta_{surf}$, units $sr^{-1}$) is estimated from the surface return signal and related to the wave-slope variance ($\sigma^2$), as detailed in Josset et al. (2010b), through..."

When considering the authors' wind speed difference statistics, I kept wondering about wind speed variations over time at a fixed point. HSRL biases relative to the dropsondes are given as

either $0.15 \pm 1.80$ m/s or $0.62 \pm 1.70$ m/s, depending on the model used. But the temporal offset between matched HSRL and dropsonde wind speed estimates can be as large as 15 minutes. How do these bias magnitudes compare to the natural variations in wind speed that would be measured at a fixed point over a 15-minute time interval? Perhaps wind speed variability information is readily available from the NOAA's National Data Buoy Center? A box and whisker plot showing wind speed differences as a function of temporal offset between the two data sets might also shed some light on this issue.

Thank you for this observation. We initially attempted to address this in the SI file, where Fig. S2 shows that there is a weak correlation between surface wind speed deltas and time. We have looked at the variations from the surface return and the optical depth calculation that will drive variations in surface wind speed and shown those statistics/plots in your other comment about providing an overview of the primary sources of uncertainty associated with calculating $\beta_{surf}$ using equation (15). This, along with the uncertainty in the dropsonde data, do not match the variability from the comparisons observed. Therefore, there is a significant potential variation of $\sim 1$ m s$^{-1}$ from the spatiotemporal differences. With this type of comparison, which involves single points from the dropsondes, we cannot perform a comparison over time to look at the variability unfortunately. Therefore, no change is made to the paper for this comment.

On lines 51–52, immediately after describing the rudiments of Hu et al., 2008 derivation, the authors introduce equation (1) by say, "The wind speed (U) was then approximated from the waveslope variance ($\sigma_2$) through this linear relationship". What I was expecting to see were the Hu equations subsequently given as equations (3.1) through (3.3). Instead, equation (1) is Cox-Munk. This section of the text (lines 44–52) should be rewritten to clearly distinguish between the original Cox-Munk equation and the subsequent CALIPSO derivation by Hu.

We apologize for this confusion. We have rewritten this section by introducing lidar retrievals in general and how Cox and Munk first related surface wind speed and surface reflectance. Then, we introduce CALIPSO afterwards and transition into the attenuation discussion.

These lines now read:

"...Therefore, instruments such as lidar are used to provide accurate surface wind speed measurements in various geographical locations to improve estimations of the MABL state globally. For instance, satellite lidar systems that measure aerosol and cloud vertical distributions, such as the lidar on board the NASA Cloud-Aerosol Lidar and Infrared Pathfinder Observation (CALIPSO) satellite, also have the capability to provide horizontally-resolved surface wind speed data. The underlying principle of lidar surface wind speed retrievals was first derived by Cox and Munk (1954), where bidirectional reflectance measurements of sea-surface glint are used to establish a Gaussian relationship between surface wind speeds and the distribution of wind-driven wave slopes. To probe these surface wave slopes, lidar instruments emit laser pulses into the atmosphere and measure the reflectance (or backscatter) of those laser pulses from particles, molecules, and the ocean surface. The magnitude of the measured signal is then used to estimate

the variance of the wave-slope distribution (i.e., wave-slope variance) and therefore surface wind speed. Note that reflectance and backscatter are used interchangeably throughout this paper.

Although many studies have expanded upon the original Cox-Munk relationship (e.g., Hu et al., 2008; Josset et al., 2008; Josset et al., 2010a; Kiliyanpilakkil and Meskhidze, 2011; Nair and Rajeev, 2014; Murphy and Hu, 2021; Sun et al., 2023), these parameterizations do not account for atmospheric attenuation by aerosols and therefore have difficulty in calibrating the measured ocean surface reflectance accurately..."

From a quick glance at the studies cited on line 54, I do not find any support for the assertion that "CALIPSO retrievals of surface wind speeds have been used in many studies". Josset et al., 2010a used AMSR winds to investigate "the normalized scattering cross section" of the CALIPSO lidar and the CloudSat radar. Kiliyanpilakkil and Meskhidze used AMSR winds and aerosol optical properties derived from CALIPSO. Nair & Rajeev used QuickScat winds and CALIPSO cloud heights. Sun et al. uses "numerical weather prediction wind vector assimilated with observed wind component" obtained from ALADIN.

Thank you for pointing this out. The goal of this sentence was to show that many studies have looked at the relationship between surface reflectance and surface wind speed since the original Cox and Munk formulation. However, this message was not communicated properly. Ultimately, we introduce these studies to later establish that the HSRL-2 can account for atmospheric attenuation by aerosols without making assumptions and therefore get an accurate measure of surface reflectance (and therefore, surface wind speed).

Revised: "Although many studies have expanded upon the original Cox-Munk relationship (e.g., Hu et al., 2008; Josset et al., 2008; Josset et al., 2010a; Kiliyanpilakkil and Meskhidze, 2011; Nair and Rajeev, 2014; Murphy and Hu, 2021; Sun et al., 2023), these parameterizations do not account for atmospheric attenuation by aerosols and therefore have difficulty in calibrating the measured ocean surface reflectance accurately..."

Figure 1 and its supporting description are all very nicely done. I commend the authors for their clear and informative presentation of this material.

Thank you for this nice comment as we spent considerable time highlighting the vertical information from the lidar, which is currently not well represented in the literature.

Please provide an overview of the primary sources of uncertainty associated with calculating $\beta_{surf}$ using equation (15). How is $\Delta\beta_{surf}$ estimated? In practice, is there some maximum $\Delta\beta_{surf} / \beta_{surf}$ above which the retrieval is deemed too unreliable for subsequent wind speed estimation?

We have provided a discussion of the uncertainties in the backscatter reflected radiances in this discussion, but the intent of the manuscript was to show the methods of the measurement approach and then perform a direct comparison with the dropsondes to assess performance. Currently, we

have not done an end-to-end assessment of the errors using the SNR from the lidar signals directly but desire to do so in the future. However, we can look at the variance of the different components of the calculation for the ocean surface backscattered radiances in Eq. 15. Based solely on detector shot-noise, we expect the calculation of the two-way optical depth (Eq. 10) to dominate the noise as this signal is much smaller as shown in Fig. 1. The variance in the ocean subsurface is small relative to the integrated surface return, so we expect the random errors from this correction to be negligible. We estimated the variability of both the integrated surface return (numerator in Eq. 15) and the optical depth term from the molecular channel (denominator) over 10 seconds. The following fractional uncertainties were calculated using all ACTIVATE data points. These are the estimated fractional uncertainties, in percent, for the 0.5 s data records.

First, estimated fractional uncertainties are shown below for 0.5 s averaged data based on variances calculated over a 10 s window (all in %):

Surface Area

- Mean: 12
- Median: 11
- 5% quartile: 6.3
- 95% quartile: 22

Optical Depth (normalization)

- Mean: 5.6
- Median: 4.5
- 5% quartile: 3.0
- 95% quartile: 11

Backscatter Reflected Radiance Uncertainty (added in quadrature)

- Mean: 13
- Median: 12
- 5% quartile: 7.8
- 95% quartile: 24

Then, estimated fractional uncertainty for 10 second average based on uncertainties above are shown. This is the averaging interval used for the comparisons of the lidar to the dropsondes winds.

Backscatter Reflected Radiance

- Mean: 3.0
- Median: 2.6
- 5% quartile: 1.7
- 95% quartile: 5.3

Next, the estimated error in wind speeds from the estimated 10 s average backscatter reflected radiance uncertainty based on the linear Cox-Munk empirical model related wave slope variance to wind speed is shown in the plots below.

[Figure]

Surface wind speed uncertainty estimated from the backscatter reflected radiance uncertainty.

[Figure]

Same as above but showing the fractional surface wind speed uncertainty.

Discussion: The mean estimated fractional error for wind speeds is less than 4-6% based on this assessment. Over the range of measurements collected for ACTIVATE, the average estimated wind speed uncertainty is less than $0.6 \text{ m s}^{-1}$ and is less than $1 \text{ m s}^{-1}$ for the 95th quartile. As pointed out by the reviewer, there are also potential bias errors due to the effects of whitecaps or foam on the water surface that are expected to affect the upper end of the measured wind speed distribution. In addition, there can be bias errors due to the assumed isotropic wave-slope variances and the assumed Gaussian distribution.

The intent of this manuscript was largely to provide an overall assessment of the retrieved wind speeds using correlative measurements from an accepted measurement (i.e., dropsondes) to quantify errors (which might be an upper limit) and to demonstrate the performance of using the high vertical resolution High Spectral Resolution lidar technique to account for both the atmospheric optical transmission and ocean subsurface scattering. The comment also asks if there is a limit of conditions that this is acceptable, and the answer is yes. We have screened data at the 0.5 s fundamental averaging interval when water clouds are detected and limited the conditions on the aircraft roll and pitch. The cloud screening is implemented for multiple reasons that include removing highly attenuated signals, which can make determining the surface location uncertain in addition to the lower signal levels. The limits on the pitch and roll were done to prevent timing delays in the incident angle of the laser beam with the surface, which can rapidly change during turns. No limits were made on the calculated optical depth for the ACTIVATE data other than screening for water clouds. We have provided the 5% and 95% quartiles to show that the signal to noise from cloud free regions is high for the ACTIVATE conditions.

Figures 4–7: ordinary least squares problems can be extremely sensitive to large outliers. Did the authors consider applying an outlier rejection scheme (e.g., Tukey fencing) before computing the regression lines shown in these figures?

Thank you for bringing this up. Initially, we only removed points that were outside the 30 km and 15 min collocation constraints. Based on your question, we've applied Tukey fencing using the standard $k = 1.5$ and see that most of the surface wind speeds above 13.3 m s$^{-1}$ would be eliminated. However, we decided to leave the high wind speed values in the analysis but recognize that the number of comparisons are limited and we point out that further measurements at these higher winds speeds are needed to fully assess both the empirical relationships and also the contribution due to whitecaps in the lidar backscatter signal. Therefore, no changes are made to the paper.

Lines 470–477: I am totally bewildered by the authors' data screening criteria; i.e., "if a wind speed retrieval is taken in an area with a high cloud fraction [...] the retrieval is deemed a missing value". Please explain what is meant by "cloud fraction" in this context. Is this vertical cloud fraction within an individual HSRL profile? Perhaps naively, I would think that (a) wind speed retrievals would be possible any time the ocean surface was reliably detected and (b) a much better QA metric could be derived from the quality of the surface backscatter signal.

Thank you for this important comment. It inspired us to restructure the case study section by removing the 11 January 2022 research study and expanding on the HSRL-2's retrieval capabilities using Research Flight 29 on 28 August 2020 and Research Flight 14 on 1 March 2020 (Sect. 3.1 now). You are correct that surface wind speed retrievals would be possible any time the ocean surface was reliably detected, which is what the 1 March 2020 discussion communicates now. The HSRL-2 can detect the surface in the breaks/gaps between clouds, so the retrievals still reliably provide data on days with broken cloud scenes. Therefore, we are not constrained to cloud-free conditions like in Hu et al. (2008), which is a significant point to highlight. However, a later comment mentions that there are limitations where the retrievals are no longer applicable such as in the cases of overcast clouds or dense fog.

Revised:

"...

[Figure]

**Figure 5: a) Flight map of the King Air (red line), Falcon (yellow line), and dropsondes (dark yellow circles) overlaid onto Geostationary Operational Environmental Satellite (GOES-16) cloud imagery for Research Flight 14 on 1 March 2020. Blue stars represent time stamps where the King Air crosses over from cloud-free to cloudy areas. b) Time series of surface wind speed data from HSRL-2 and dropsondes for the same flight, where lines signify total HSRL-2 surface wind speed data and circles indicate collocated surface wind speed data points. Blue dashed lines represent time stamps of interest as indicated in a).**

As the aircraft approaches the cloud scene at 19:18, there is a noticeable and steady increase of HSRL-2 surface wind speeds. The reverse observation is seen when the aircraft approaches 21:15, where the HSRL-2 surface wind speeds start to decrease steadily. As highlighted in the 28 August 2020 case study, the high horizontal spatial resolution of the HSRL-2 retrievals enables these spatial gradients to be observed. Another important takeaway is the HSRL-2 is still able to sample the surface in cloud scenes, as seen by the almost complete surface wind speed profile in Fig. 5b. Although a gap in data occurs at 20:15 where cloud cover is most substantial, some retrievals are still present in that area. The reason is that the HSRL-2 can probe the surface through gaps between clouds, allowing for the surface wind speed retrievals to take place. Although the HSRL-2 retrievals would be unavailable in overcast cloud scenes, the ability of the instrument to sample the surface in broken cloud fields and not just cloud-free scenes is a significant benefit of the lidar and the HSRL technique."

Minor Remarks
The formatting of equation (1) is ambiguous. Decimal notation would be much, much better I think (e.g., 0.003 instead of 3.0E – 3).

Thank you for this comment. Reviewer 1 noted that it is uncommon to write equations in the introduction, so we removed Eq. 1 and left any discussion on wind speed – wave-slope to the Methods section (i.e., Sect. 2.4) and switched to decimal notation as suggested.

Line 105: what value did the authors use for the Fresnel coefficient? Note: Hu et al., 2009 use 0.0209, Josset et al., 2010a use 0.0213, and Venkata and Reagan 2015 use 0.0205.

Thank you for noting this omission and it follows a more general comment from the other reviewer. We have provided a paragraph on the relevant parameters of the instrument and geometry that is implemented before the methodology. Specifically, we use a constant value of 0.0205 for the Fresnel coefficient listed in Venkata and Reagan (2016) due to the limited range of angles analyzed and the constant wavelength from the lidar. The change in the text is provided below.

Line 105: what is the typical off-nadir angle for the HSRL measurements? Should readers assume nadir pointing, so that $\theta = 0$?

As noted in the previous comment, we have added an instrument description which we hope addresses the lack of information provided on the viewing geometry. Specifically, the calculations are limited to relatively small angles, but we note that it is important to account for them in the calculations. The nominal aircraft pitch would vary depending on flight conditions from 3 - 5° for the aircraft used, but the angle of incidence could be as large as 10° accounting for both the pitch and roll of the aircraft. We do not go into specific details in the text, but we did limit the roll angles to +/- 3° from the median values to prevent rapid changes in the aircraft altitude data during the data averaging interval and could have a slight lag in time (< 0.5 s). Most flights were conducted with limited turns (< 2 - 3) when over the water.

Added: "For the surface wind speed calculations, data are screened to limit the pitch and roll to less than +/- 3° from the median values, which are approximately 0° for the roll and 3° - 5° for pitch on the King Air."

Line 110: "Eq. 3.3 is  identical to the log-linear relationship proposed by Wu (1990)."

We apologize for this confusion. We originally tried to communicate that Hu did not simply reuse the Cox-Munk and Wu relationships and these identical results came from his own derivation.

Now, the line reads: "The relationships shown in Eqs. 3.1 – 3.3 were derived by Hu using the comparisons between AMSR-E surface wind speeds and CALIPSO backscatter reflectance mentioned in Sect. 1 and agree identically with the Cox-Munk relationship for surface wind speeds between 7 m s$^{-1}$ and 13.3 m s$^{-1}$ and the log-linear relationship proposed by Wu (1990) for surface wind speeds above 13.3 m s$^{-1}$."

Line 111: change "to be" to "being"

This line is now removed because we realize it is misleading. The Venkata-Reagan model is also used in CALIPSO retrievals.

Line 192: practically speaking, is there some maximum AOD above which surface wind speeds are not considered reliable? Or are there perhaps some meteorological conditions in which the method is not applicable (e.g., exceptionally dense surface-hugging fogs)?

You are correct and we comment on this in the discussion of the uncertainty above. In general, for cloud-free conditions, the retrievals are possible as noted in the 1 March 2020 case study. We screen for clouds based on the atmospheric backscatter measurements and do not perform the retrievals as there are a lot of conditions that make this challenging including even finding the surface. We agree that dense fog would be one of these conditions and that would be excluded due to the screening process.

Added: "Although the HSRL-2 retrievals would be unavailable in overcast cloud scenes, the ability of the instrument to sample the surface in broken cloud fields and not just aerosol- and cloud-free scenes is a significant benefit of the lidar and the HSRL technique."

Figure 3: use different line colors and/or line types to plot the two different sets of HSRL wind speed retrievals.

We removed most Cox-Munk comparisons throughout the paper, so this figure (now Fig. 4b) reflects this change.

Lines 299–300: In the figure caption, the authors say, "A few collocated Hu08 and CM54 wind speed data points are on top of each other owing to similar values." They could (and should) eliminate any ambiguity by specifying UTC for these pairs of points.

As mentioned in the previous reply, most Cox-Munk comparisons were removed from most of the paper. Therefore, this line has been removed.

Lines 319–325: I would have appreciated a bit more detail here. The authors' description does not provide sufficient information to distinguish the bisector method from other 'errors in variables' techniques (e.g., Deming regression and orthogonal distance regression). Is the bisector method especially effective for problems of this sort? Or will any errors in variables method do equally well?

We appreciate you bringing up this point. This section has been expanded in Sect. 2.5, but we will provide an explanation here as well. Although OLS-bisector, Deming, and orthogonal distance regressions are all errors-in-variable techniques as you mentioned, we have reason to believe that OLS-bisector is the better choice for this study. Orthogonal distance regression assumes that the total error in Y is equal to the total error in X, which is probably not true considering the wind speed data are coming from two different instruments. Deming regression assumes that the

measurement error ratio between X and Y is constant (default of 1), so one must provide their own ratio before performing the regression (which is not necessarily straightforward) (Wu and Yu, 2018).

Least squares bisector is where Y is regressed on X (standard OLS) and then X is regressed on Y (inverse OLS) (Ricker, 1973). Then, the technique minimizes the distance of the observation points by drawing a line that bisects the angle of the two regression lines. Then, the variance and covariance of this line are calculated to provide a measure of error in both X and Y. We do need to assume that the relationship between X and Y is approximately linear, but we prefer this method because it is straightforward to use and can calculate the error present in both data sets without making a priori assumptions like in orthogonal distance or Deming.

Added: "Since OLS-bisector is less common than linear regression, a brief explanation of their differences is provided. In linear regression, X is treated as the independent variable while Y is treated as the dependent variable. In other words, one observes how Y varies with changes to fixed X values. OLS-bisector is known as an errors-in-variable regression technique, where X and Y are both dependent variables and thus both subject to error. OLS-bisector regresses Y on X (standard OLS) and then regresses X on Y (inverse OLS), then bisects the angle of these two regression lines (Ricker, 1973). Although other errors-in-variable techniques exist (e.g., Deming regression, orthogonal distance regression), OLS-bisector is chosen because it calculates the error present in both data sets using the bisector rather than assuming an error a priori like the examples mentioned (Wu and Yu, 2018)."

References

Ricker, W. E.: Linear Regressions in Fishery Research, Journal of the Fisheries Research Board of Canada, 30, 409-434, 10.1139/f73-072, 1973.

Wu, C. and Yu, J. Z.: Evaluation of linear regression techniques for atmospheric applications: the importance of appropriate weighting, Atmos. Meas. Tech., 11, 1233-1250, 10.5194/amt-11-1233-2018, 2018.

One Reviewer's Opinion
A scatter plot of dropsonde wind speeds (U) versus matching values of $\sigma_2$ (derived using equation (3) and $\beta_{surf}$ computed using equation (15)) would have made this paper enormously more interesting. Both Cox-Munk and Hu et al., 2008 are approximations of the true relationship between wave slope variance and surface wind speeds. The collocated measurements reported in this manuscript offer a superb opportunity to evaluate the relative merits of both models. Perhaps this tantalizing topic can be briefly explored in an appendix included in a revision to the current manuscript.

Thank you for this great advice. This inspired us to restructure the Results section. We first introduce Fig. 7 (in wind speed space) to show that the Cox-Munk model overestimates the dropsonde wind speeds more so than the Hu model for winds below 7 m s$^{-1}$. Then, we add the plot

you suggest of HSRL-2 wave-slope variance versus dropsonde surface wind speeds (now Fig. 8) to better compare the merits of both models. When comparing the HSRL-2 measurements of wave-slope variance to the wave-slope variance computed using the dropsonde surface wind speeds, we find that the Hu et al. (2008) model provides a better representation of wave-slope variance than the more commonly used Cox-Munk model.

Added: "Now that the HSRL-2 retrievals have been broadly evaluated, Fig. 7 shows how their accuracy varies per 1 m s$^{-1}$ interval in surface wind speed. This plot also provides the opportunity to compare the Hu et al. (2008) model with the models proposed by Cox and Munk (1954) and Wu (1990) to see if some of the error in the HSRL-2 retrievals can be attributed to model characteristics.

[Figure]

**Figure 7: HSRL-2 surface wind speed using Hu, Cox-Munk, and Wu models versus mean dropsonde surface wind speed calculated per 1 m s$^{-1}$ bin. A histogram of dropsonde surface wind speeds is also included to show their distribution.**

It is seen that the mean Cox-Munk and Wu surface wind speed values are higher than the mean Hu values from 0 m s$^{-1}$ to 7 m s$^{-1}$, showing that the Cox-Munk and Wu relationships overestimate dropsonde surface wind speeds more than the Hu relationship. The variability (i.e., STD) around the mean per bin is similar between the three models, which is 1.59 m s$^{-1}$ for Hu, 1.43 m s$^{-1}$ for Cox-Munk, and 1.55 m s$^{-1}$ for Wu on average. Although similar, the STD of the Hu surface wind speeds found here is ~0.4 m s$^{-1}$ lower than the one found in Fig. 6. This could be attributed to an STD not being able to be calculated for the 17 to 18 m s$^{-1}$ bin since it only contained one point.

Although it is apparent Cox-Munk and Wu retrievals overestimate dropsonde observations for surface wind speeds below 7 m s⁻¹, it is still unclear which of the models perform better overall. Therefore, the y-axis from Fig. 7 is converted to wave-slope space and the result of this modification is shown in Fig. 8. HSRL-2 wave-slope is used because it directly reports the original measurements of surface reflectance rather than estimated values of surface wind speed. Using the original data ensures that uncertainty is coming from the actual HSRL-2 – dropsonde comparisons rather than from potential errors in the conversion from wave-slope to surface wind speed.

[Figure]

**Figure 8: HSRL-2 wave-slope variance versus mean dropsonde surface wind speed calculated per 1 m s⁻¹ bin. Ideal Hu, Cox-Munk, and Wu distributions are included to show how well observed dropsonde data match with each parameterization. A histogram of dropsonde surface wind speeds is also included to show their distribution.**

From Fig. 8, it is more easily seen how the dropsonde surface wind speed distribution compares with Hu, Cox-Munk, and Wu parameterizations. Dropsonde surface wind speeds match quite closely to Hu and Cox-Munk parameterizations as opposed to the Wu parameterization between 7 m s⁻¹ and 13.3 m s⁻¹, although some divergence is seen above ~10.5 m s⁻¹. However, a critical observation that is more apparent in Fig. 8 than Fig. 7 is how the dropsonde data most resemble the Hu distribution for surface wind speeds below 7 m s⁻¹. This improvement is substantial, especially since most of the surface wind speeds in ACTIVATE fall into this category. Surface wind speeds above 13.3 m s⁻¹ substantially diverge from all models, especially above 16 m s⁻¹. As mentioned previously, there are few surface wind speed observations in this category, so more measurements are necessary to make meaningful comparisons between the two data sets. Overall,

Figs. 7 and 8 demonstrate the benefits of using the Hu parameterization in this study and why surface wind speeds above 13.3 m s$^{-1}$ are not the main focus of the comparisons in this section. Further analysis is warranted to rigorously compare the performance of various surface reflectance models and potentially apply corrections (i.e., whitecap correction for surface wind speeds above 13.3 m s$^{-1}$), but the aim of this paper is to evaluate LARC's HSRL-2 surface wind speed retrieval algorithm using the available ground-truth dropsonde measurements."

**References**

Cox and Munk, 1954: Measurement of the Roughness of the Sea Surface from Photographs of the
Sun's Glitter, *J. Opt. Soc. Am.*, **44**, 838–850, https://doi.org/10.1364/JOSA.44.000838.
Hu et al., 2008: Sea surface wind speed estimation from space-based lidar measurements, *Atmos. Chem. Phys.*, **8**, 3593–3601, https://doi.org/10.5194/acp-8-3593-2008.
Josset et al., 2010a: Multi-Instrument Calibration Method Based on a Multiwavelength Ocean Surface Model, *IEEE Geosci. Remote Sens. Lett.*, **7**, 195–199, https://doi.org/10.1109/LGRS.2009.2030906.
Josset et al., 2010b: Lidar equation for ocean surface and subsurface, *Opt. Express*, **18**, 20862–20875, https://doi.org/10.1364/OE.18.020862.
Lancaster et al., 2005: Laser pulse reflectance of the ocean surface from the GLAS satellite lidar, *Geophys. Res. Lett.*, **32**, L22S10, https://doi.org/10.1029/2005GL023732.
Venkata and Reagan, 2016: Aerosol Retrievals from CALIPSO Lidar Ocean Surface Returns, *Remote Sens.*, **8**, 1006, https://doi.org/10.3390/rs8121006.

---

## Author Response (AR2)

We thank the reviewer for their thorough review of this manuscript and bringing up an important detail about the nature of the HSRL-2 viewing geometry. We provide a response to this comment in blue font.

Reviewer comment:

The authors have addressed all my comments and provided clear explanations or amendments to previously overlooked aspects. This study can be accepted with following minor remarks applied. Once the authors have provided clarified version of the methodology, it becomes clear that they are applying surface reflectance-wind parametrizations for a nadir or a near-nadir system, but it is not clearly articulated.

Comment about difference between nadir and non-nadir lidar systems to the authors: If you use nadir system, in this case, this (1) should be clearly stated in the methodology. What is the incidence angle of their HSRL system? Also, if it's nadir or near nadir, (2) the introduction and methodology should shortly mention that this logical rationale you provide through methodology and results holds true for near-nadir and nadir systems. I think a reader should be clearly informed that you are talking about near nadir or nadir system and a minor red flag should be raised over the non-nadir systems. My formulation is as following, but you can include this rationale in any form in your study. "At nadir and near-nadir incidence angles, most contribution of lidar surface signal comes from ocean surface [these Josset et al. studies that you mentioned], which makes it possible to introduce relatively simplified models of sea surface reflectance. However, Li et al. [2010] had demonstrated that at the higher incidence angle lidar systems (> 15o), the sensitivity of lidar backscatter signal from ocean surface would rapidly decrease at such highly non nadir incidences being shifted towards subsurface contribution. More recent lidar study based on highly non-nadir (~37o) Aeolus UV HSRL lidar [Labzovskii et al., 2023] indirectly confirmed this phenomenon by showing low agreement between passive remote sensing reflectivity and Aeolus surface reflectivity parameters particularly only over water surfaces such as oceans. Thus, an opportunity to retrieve ocean surface winds using lidar ocean backscattering has been shown effective only for nadir or near-nadir lidar systems such as x incidence angle of HSRL system we analyze." There are more studies on this topic, but these two references including our recently published study on this should be enough to raise flag about non-nadir HSRL lidar systems for a reader. It's up to you to include these comments or not though.

Li et al. [2010] - DOI: 10.1175/2009JTECHA1302.1
Labzovskii et al. [2023] - https://doi.org/10.1038/s41598-023-44525-5

Response:

Thank you for bringing up this important point. We agree that the HSRL-2 nadir-viewing geometry should have been more carefully explained and how this affects the use of surface

reflectance – wind speed parameterizations. We have made the following additions to articulate these points.

Abstract:

> Original: "The HSRL-2 can directly measure vertically resolved aerosol backscatter and extinction profiles without additional constraints or assumptions, enabling the instrument to accurately derive atmospheric attenuation and directly determine surface reflectance (i.e., surface backscatter)."

> Modified: "The HSRL-2 is a nadir-viewing lidar that can directly measure vertically resolved aerosol backscatter and extinction profiles without additional constraints or assumptions, enabling the instrument to accurately derive atmospheric attenuation and directly determine surface reflectance (i.e., surface backscatter)."

Introduction:

> Added: "Note that the HSRL-2 operates at a nadir-viewing geometry, which is detailed more in Sect. 2.4. At nadir or near-nadir incidence angles, the surface contribution of the lidar surface backscatter signal is the largest and is therefore sensitive to changes in wind speed (Josset et al., 2008; Josset et al., 2010a; Josset et al., 2010b), making it possible to introduce relatively simplified models of sea surface reflectance. However, Li et al. (2010) demonstrated that at the higher incidence angle lidar systems (> 15°), the sensitivity of the lidar surface signal would rapidly decrease as these highly non-nadir incidences shift the signal towards a subsurface contribution rather than a surface one. A more recent lidar study based on the highly non-nadir (~37°) Aeolus UV HSRL lidar (Labzovskii et al., 2023) indirectly confirms this phenomenon by showing low agreement between passive remote sensing reflectivity and Aeolus surface reflectivity parameters over water surfaces such as oceans. For these reasons, an opportunity to retrieve ocean surface wind speeds using lidar ocean backscattering has been shown to be effective only for nadir or near-nadir lidar systems such as the HSRL-2."

Methods (Sect. 2.4):

> Added: "As noted in the Introduction, the HSRL-2 is operated in a nadir-only viewing geometry (i.e., not scanning). However, there is a small offset from this nadir incidence angle due to the pitch and roll angles of the King Air aircraft. This offset angle is measured by the Applanix INS and is then used in Eq. 2 to derive the wave-slope variance. The median pitch and roll angles depend on the flight conditions (e.g., wind and fuel loads), but ranged from 2 - 5° for pitch and < 1° for roll during ACTIVATE flights. The surface wind speed data are screened to limit the pitch and roll to less than ± 3° from the median values, resulting in HSRL-2 incidence angles of < 3° for roll and < 8° for pitch. This screening effectively selects cases where the aircraft is flying straight and level legs."

---

## Author Response (AR3)

We thank the editor for suggesting final technical corrections in the abstract and conclusion. The corrections are found in blue font below.

Editor suggestion:

The clarifications in the introduction and body of the text are very clear, while further main clarification in the abstract and conclusions will be useful for clarity. For example, "This study introduces the High Spectral Resolution Lidar" could be changed in "This study introduces the nadir High Spectral Resolution Lidar".

Response:

Thank you for bringing this up. We have included the mention of "nadir" in the lines below.

Abstract:

> Original: "To provide these measurements, this study introduces and evaluates a new surface wind speed data product from NASA Langley Research Center's High Spectral Resolution Lidar – generation 2 (HSRL-2) using data collected as part of NASA's Aerosol Cloud meTeorology Interactions oVer the western ATlantic Experiment (ACTIVATE) mission."

> Modified: "To provide these measurements, this study introduces and evaluates a new surface wind speed data product from NASA Langley Research Center's **nadir-viewing** High Spectral Resolution Lidar – generation 2 (HSRL-2) using data collected as part of NASA's Aerosol Cloud meTeorology Interactions oVer the western ATlantic Experiment (ACTIVATE) mission."

Conclusion:

> Original: "This study introduces the High Spectral Resolution Lidar – generation 2 (HSRL-2) surface wind speed retrieval method, demonstrates its use, and evaluates its accuracy using NCAR AVAPS dropsonde data collected during the NASA ACTIVATE field campaign."

> Modified: "This study introduces a new 10 m surface wind speed product from NASA Langley Research Center's (LaRC's) **nadir-viewing** High Spectral Resolution Lidar – generation 2 (HSRL-2) instrument and demonstrates its use and accuracy. The HSRL-2 retrievals are evaluated using NCAR AVAPS dropsonde surface wind speed data collected during the NASA ACTIVATE field campaign."